# DOCKSMITH: Scaling Reliable Coding Environments via an Agentic Docker Builder

**Jiaran Zhang** [* 1]   **Lu Ma** [* 1]   **Yanhao Li** [2]   **Fanqi Wan** [1]   **Di Qi** [1]   **Xin Wu** [1]   **Zhewei Huang** [1]   **Liangyu Chen** [1]
**Yingwei Ma** [3]   **Qi Han** [1]   **Xiangyu Zhang** [1]

## Abstract

Reliable Docker-based environment construction is a dominant bottleneck for scaling execution-grounded training and evaluation of software engineering agents. We introduce DOCKSMITH, a specialized agentic Docker builder designed to address this challenge. DOCKSMITH treats environment construction not only as a preprocessing step, but as a core agentic capability that exercises long-horizon tool use, dependency reasoning, and failure recovery, yielding supervision that transfers beyond Docker building itself. DOCKSMITH is trained on large-scale, execution-grounded Docker-building trajectories produced by a SWE-Factory–style augmented with a loop-detection controller and a cross-task success memory. Training a 30B-A3B model on these trajectories achieves open-source state-of-the-art performance on Multi-Docker-Eval, with *39.72% Fail-to-Pass* and *58.28% Commit Rate*. Moreover, DOCKSMITH improves out-of-distribution performance on SWE-bench Verified, SWE-bench Multilingual, and Terminal-Bench 2.0, demonstrating the broader agentic benefits of environment construction. Our model and Docker-building trajectories are publicly available at here.

## 1. Introduction

Recent advances in software engineering agents increasingly rely on scaling *execution-grounded* supervision, typically collected through automated pipelines that mine historical code changes and execute repository tests to obtain executable feedback signals (Wang et al., 2025; Liu et al., 2024b; Ma et al., 2025b). A central requirement of these pipelines is reliable environment construction, which transforms a static repository into an executable Docker environment in which tests and diagnostics can be run (Guo et al., 2025; Yang et al., 2025; Zeng et al., 2025).

In practice, Docker-based environment setup is highly failure-prone (Yang et al., 2025; Pan et al., 2025). Heterogeneous dependencies, system-level conflicts, and undocumented build assumptions frequently cause environment construction to fail, preventing many repositories from reaching the execution and validation stages. This difficulty persists even for strong models, including Claude-4-Sonnet (Anthropic, 2025a) and Gemini-2.5-Flash (Comanici et al., 2025): on Multi-Docker-Eval, a benchmark specifically designed to evaluate Docker-based environment construction, such models struggle to reliably pass environment setup and validation (Fu et al., 2025). Consequently, the brittleness of automated environment construction sharply limits the yield of usable trajectories, leaving only a small fraction of repositories suitable for agent training.

Beyond being a practical bottleneck, we show that environment construction is a core agentic capability, on par with bug fixing and feature implementation. Rather than treating repository bootstrapping as a mere prerequisite, we cast it as a verifiable, execution-grounded task that exercises long-horizon reasoning, tool use, and failure recovery. From this perspective, environment construction provides a strong training signal that can transfer to broader agentic tasks.

Motivated by this view, we build an agentic Docker-building pipeline based on SWE-Factory (Guo et al., 2025), augmented with a loop-detection controller and cross-task reuse of verified solutions. Using this pipeline, we curate a large-scale dataset of high-quality Docker-building trajectories capturing fine-grained commands, configurations, diagnostics, and recovery steps. Training on this data yields DOCKSMITH, a dedicated 30B-A3B Docker-building model that achieves open-source state-of-the-art performance on Multi-Docker-Eval, substantially improving Docker build success at a relatively modest model scale.

---
[*]Equal contribution [1]StepFun, Beijing, China [2]Peking University, Beijing, China [3]Hong Kong University of Science and Technology, Hong Kong. Correspondence to: Qi Han <hanqer@stepfun.com>, Xiangyu Zhang <robert.zhang@stepfun.com>. The work was supported by the National Science and Technology Major Project of China (2023ZD0121300).

*Proceedings of the $43^{rd}$ International Conference on Machine Learning*, Seoul, South Korea. PMLR 306, 2026. Copyright 2026 by the author(s).

DOCKSMITH delivers a large improvement in Docker-based environment construction. On Multi-Docker-Eval, DOCK-SMITH achieves 39.72% Fail-to-Pass and a 58.28% Commit Rate, setting open-source state of the art and surpassing the previously reported 37.7% F2P upper bound. This gain is accompanied by markedly more stable build-and-repair behavior, with substantial reductions in Dockerfile, eval-script, and diagnostic errors. Beyond environment setup, DOCK-SMITH's training signal transfers to general agentic software engineering: joint training with Docker-building trajectories improves SWE-bench performance and yields up to 3.37 points on interaction-heavy tasks such as Terminal-Bench. DOCKSMITH strengthens executable-environment construction while also improving dependency reasoning, execution planning, and failure recovery.

Successfully bootstrapped environments provide more than a working code base: they enable scalable data curation and learning through reliable execution and verification. Deterministic execution feedback and verifiable unit tests enable iterative, interactive coding, supporting the generation of high-quality synthetic data and reward signals. Leveraging DOCKSMITH, we curate over 30k verified environments spanning more than 15k GitHub repositories and more than 20 programming languages, establishing a robust foundation for training and aligning generalist code agents.

Overall, our results position environment construction not as a mere preprocessing step, but as a core agentic task with transferable supervision. DOCKSMITH serves both as a practical mechanism for scaling execution-grounded code agent pipelines and as empirical evidence that explicitly modeling environment setup can advance autonomous software engineering.

## 2. Method

We present an end-to-end framework for training Docker-building agents from real-world software development data, as shown in Figure 1. Leveraging large-scale GitHub repositories, we curate a high-quality dataset of accepted, test-backed pull requests and transform them into agentic environment-construction trajectories. Extending the SWE-Factory pipeline, we introduce a multi-agent system augmented with loop-aware, cross-task memory to reliably generate Docker-building rollouts. We then train DOCK-SMITH via fine-tuning on filtered and balanced trajectories, using curriculum sampling and joint training with general coding data to improve robustness and transferability.

### 2.1. Data Sourcing

We derive our training set from real-world GitHub repositories. The pipeline comprises three phases: repository selection, pull request filtering, and quality enhancement.

**Repository Selection.** We prioritize high-impact projects based on community validation and activity. Repositories are included if they possess $\geq 500$ stars and $\geq 200$ forks. We target 10 major programming languages: Python, PHP, TypeScript, Go, C++, JavaScript, Java, Rust, C, and Ruby—yielding over 15,000 repositories, providing a diverse base for cross-language generalization.

**Pull Request Crawling.** We crawl pull requests as problem–solution pairs and retain only merged PRs with substantive code changes and test-related modifications, ensuring both human approval and executable validation signals. The resulting PRs provide reliable supervision for training. To avoid benchmark contamination, we apply repository-level deduplication against all evaluation benchmarks.

**Quality Enhancement.** To improve the clarity and completeness of problem specifications, we employ language models to refine brief or ambiguous PR descriptions. This process expands and clarifies the problem statements while preserving their original technical intent and constraints. The resulting dataset comprises approximately 200,000 high-quality, curated instances. After processing through the agentic Docker-building pipeline (Section 2.2), the resulting trajectory training corpus contains 692k trajectory steps spanning 45k unique environment construction instances drawn from 19k GitHub repositories. To ensure broad generalization, the language distribution is anchored by major ecosystems: TypeScript (21.32%), JavaScript (20.25%), Go (18.76%), Java (8.63%), PHP (6.87%), and Python (5.17%), while maintaining a robust long-tail of system and compiled languages including Rust, C++, and Ruby.

### 2.2. Agentic Docker-Building Pipeline

We extend the environment construction pipeline introduced in SWE-Factory, which models Docker-based repository setup as an iterative, execution-grounded refinement process. Figure 1 depicts the SWE-Factory control flow and highlights our two extensions: (i) a *loop-detection* controller that explicitly guards against stalled, repetitive multi-agent repair, and (ii) a *cross-task* variant of its success-memory retrieval that enables reuse beyond the current run.

**Specialized agents.** The pipeline orchestrates four LLM-based agents that exchange artifacts and structured feedback: (i) the *Context Retrieval Agent*, which inspects the repository via iterative tool use to collect environment-relevant signals such as dependency manifests, build scripts, CI configurations, and test entry points; (ii) the *Dockerfile Agent*, which synthesizes or patches a Dockerfile based on retrieved context and execution feedback; (iii) the *Eval Script Agent*, which generates an executable script to configure the container workspace and invoke the target test commands; and (iv) the *Test Analysis Agent*, which executes the Docker build and test pipeline and distills raw logs into structured

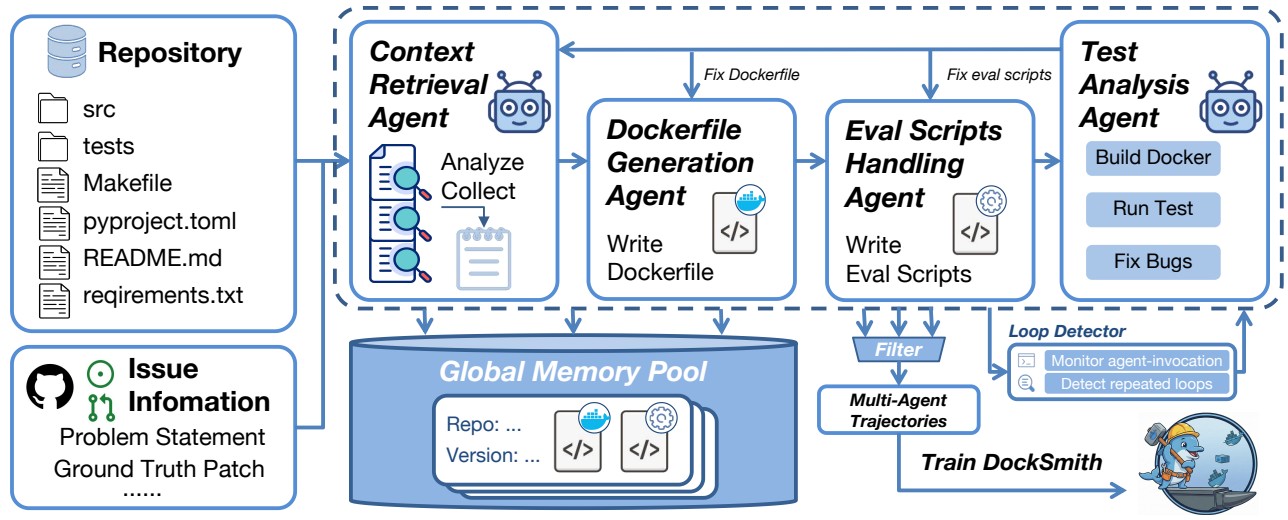

*Figure 1.* DOCKSMITH training framework. A multi-agent pipeline with four agents (Context Retrieval, Dockerfile, Eval Script, Test Analysis) generates execution-grounded Docker-building trajectories from test-backed pull requests in an iterative repair loop. Loop detection and cross-task success memory are used, followed by trajectory filtering and curriculum sampling for supervised fine-tuning.

failure summaries for the next repair iteration.

**Loop detection and intervention.** In practice, iterative multi-agent repair may devolve into a repetitive failure mode where the same subset of agents is invoked without measurable progress. We introduce a loop-detection controller that monitors recent agent-invocation traces alongside execution failure signatures. When repeated activation of an identical agent combination fails to improve outcomes for several rounds, the controller intervenes by enforcing diversification—such as activating alternative agents or strategies—to break deadlocks and reduce wasted iterations.

**Cross-task memory pooling.** SWE-Factory maintains a *memory pool* of validated environment solutions. We augment this mechanism to support cross-task retrieval, enabling previously verified (Dockerfile, eval script) pairs to serve as lightweight demonstrations for new repositories.

### 2.3. Model Training

We train DOCKSMITH via supervised fine-tuning on execution-grounded Docker-building trajectories. Each instance is a multi-turn interaction trace that includes Dockerfile edits, tool calls, build and test commands, execution logs, and repair actions. Since raw agent rollouts are noisy and often characterized by redundant exploration, we construct a compact, high-signal training corpus through a three-stage curation process. This process balances data across programming languages, filters and compresses trajectories to remove low-information steps, applies a complexity-aware curriculum to emphasize increasingly challenging builds. To retain general software engineering capability without sacrificing Docker specialization, we perform joint training

with general coding trajectories, enabling DOCKSMITH to better integrate environment construction with downstream code understanding and execution.

#### 2.3.1. DATA BALANCING AND SAMPLING

**Cross-language data balancing.** Our corpus encompasses multiple programming languages, with the majority of trajectories originating from core ecosystems such as Python and C/C++, and a smaller proportion from long-tail languages including Rust and PHP. Preliminary trials indicated that indiscriminate upsampling of long-tail data could induce distribution shift and degrade overall performance. Consequently, we adopt a conservative balancing policy that preserves the core-language distribution while incorporating long-tail samples in a controlled manner. Specifically, we enforce per-language token contribution limits during corpus construction and training, so that the long-tail languages improve coverage without dominating the batch. This strategy stabilizes training and broadens language coverage while maintaining a controlled distribution shift.

**Filtering long and redundant rollouts.** Even among successful rollouts, we observe substantial redundancy resulting from repeated tool invocations and long interactions. To mitigate noise, we eliminate trajectories that exhibit unproductive exploration patterns. Specifically, we discard rollouts that repeatedly invoke the same agent module, contain excessively many turns, or accumulate an unusually large number of messages. These thresholds are intended to trim the heavy tail of rollout lengths while preserving the majority of informative trajectories.

**Complexity-based curriculum sampling.** To manage

*Table 1.* Overall Multi-Docker-Eval performance of different models on SWE-Factory.

| Model | Outcome Metrics | | Process Metrics | | |
|---|---|---|---|---|---|
| | **F2P** (%) | **Commit** (%) | **Avg input** | **Avg output** | **Avg docker** |
| *Open-source Models* | | | | | |
| DeepSeek-v3.1 | 37.72 | 52.89 | 158.11 | 17.15 | 1.02 |
| DeepSeek-R1 | 26.65 | 41.72 | 138.05 | 60.10 | 1.02 |
| GPT-OSS-20B | 17.17 | 29.44 | 184.23 | 58.37 | 0.87 |
| GPT-OSS-120B | 27.00 | 37.72 | 128.31 | 30.17 | 0.83 |
| Kimi-K2-0905 | 37.62 | 55.49 | 113.02 | 7.92 | 1.01 |
| Kimi-K2-thinking | 36.53 | 52.69 | 162.12 | 59.40 | 1.05 |
| Qwen3-235B-A22B | 23.65 | 34.53 | 101.46 | 38.87 | 1.00 |
| *Closed-source Models* | | | | | |
| Claude-Sonnet-4 | 35.53 | 47.41 | 182.85 | 15.01 | 1.17 |
| GPT-5-Mini | 34.13 | 49.60 | 339.94 | 103.32 | 0.95 |
| Gemini-2.5-Flash | 29.44 | 40.62 | 153.43 | 32.60 | 0.97 |
| *Our Method* | | | | | |
| Qwen3-Coder-30B-A3B-Instruct | 19.46 | 34.13 | 150.10 | 13.75 | 0.93 |
| DOCKSMITH | **39.72** | **58.28** | 207.68 | 26.38 | 1.13 |

dataset size while maintaining a meaningful difficulty distribution, we employ a complexity-based sampling strategy. We first retain only verified-success instances to ensure reliable supervision. Subsequently, we estimate Dockerfile difficulty using lightweight structural signals, enabling data stratification and subsampling without incurring the cost of additional builds. Concretely, we define a scalar complexity score using three structural features: $L(d)$ (number of non-empty Dockerfile lines), $R(d)$ (number of RUN instructions), and $P(d)$ (number of distinct packages installed via apt-get/apt install). Since these raw dimensions vary drastically in scale—e.g., Dockerfile length typically ranges in the hundreds while RUN count rarely exceeds a handful—we apply *parameter-free Z-score normalization* to ensure each feature contributes equally to the final score:

$$\text{Score}(d) = \frac{L(d) - \mu_L}{\sigma_L} + \frac{R(d) - \mu_R}{\sigma_R} + \frac{P(d) - \mu_P}{\sigma_P}, \quad (1)$$

where $\mu_i$ and $\sigma_i$ denote the corpus-level mean and standard deviation of feature $i$, computed over all successfully built Dockerfiles in our training pool. This normalization ensures that "Hard" trajectories derive their difficulty from dense dependency resolution rather than simply being verbose, and it obviates the need for manually tuned weighting coefficients. Using this score, we bucket instances into Easy, Medium, and Hard, and sample them with a 1:2:2 ratio to ensure sufficient exposure to challenging builds. This curriculum prevents the training set from being dominated by trivial one-shot cases and improves robustness on dependency-heavy projects.

### 2.3.2. JOINT TRAINING WITH CODING TRAJECTORIES

When training DOCKSMITH, we train Docker-building trajectories together with general SWE / Coding trajectories. The intuition is that SWE trajectories provide task-level execution and validation patterns (e.g., edit–run–debug loops) that complement environment construction, helping the model avoid over-specializing to Docker-only behaviors. We adopt *token-level mixing*: the token budget for coding trajectories is held constant, while we vary the Docker-building token budget to control the mixing ratio, keeping total compute comparable across runs. The effect of this joint-training strategy is evaluated in the ablation study in Section. 3.4.

## 3. Experiments

### 3.1. Experiment Setup

**Benchmarks.** We evaluate Docker-building performance on Multi-Docker-Eval (MDE) (Fu et al., 2025), a benchmark designed to assess repository-level environment construction across diverse real-world software projects and programming languages. Each instance requires the model to synthesize a valid Dockerfile, build the environment, and ensure that the repository can be executed or tested successfully. Success is measured by whether the build process completes without failure and whether the resulting environment supports downstream execution.

To test whether Docker-building improvements transfer beyond environment setup, we evaluate on three widely adopted agentic software-engineering benchmarks that are not used as training objectives in our setting: (1) SWE-

*Table 2.* Multi-Docker-Eval F2P (%) across programming languages by different models. Gray-shaded rows indicate closed-source models, while the remaining models are open-source.

| Model | Language | | | | | | | | |
|---|---|---|---|---|---|---|---|---|---|
| | Python | JavaScript | Java | C++ | C | Go | Ruby | Rust | PHP |
| DeepSeek-v3.1 | 47.86 | 45.83 | 14.29 | **18.89** | **33.33** | 57.50 | 51.67 | 20.83 | **42.22** |
| DeepSeek-R1 | 32.48 | 40.83 | 9.52 | 10.00 | 22.50 | 46.67 | 35.83 | 10.00 | 25.56 |
| GPT-OSS-20B | 15.38 | 17.50 | 8.57 | 2.22 | 15.00 | 46.67 | 25.00 | 9.17 | 7.78 |
| GPT-OSS-120B | 35.90 | 36.67 | 10.48 | 10.00 | 23.33 | 53.33 | 32.50 | 14.17 | 22.22 |
| Kimi-K2-0905 | 47.01 | 49.17 | 12.38 | 12.22 | 34.17 | 61.67 | 51.67 | 22.50 | 36.67 |
| Kimi-K2-thinking | 49.57 | 46.67 | 11.43 | 11.11 | 37.50 | 63.33 | 48.33 | 17.50 | 32.22 |
| Qwen3-235B-A22B | 21.37 | 37.50 | 9.52 | 5.56 | 17.50 | 51.67 | 30.00 | 15.83 | 15.56 |
| Claude-Sonnet-4 | 41.88 | 48.33 | 8.57 | 18.89 | 32.50 | 66.67 | 43.33 | 18.33 | 30.00 |
| GPT-5-Mini | 43.59 | 49.17 | 14.29 | 15.56 | 29.17 | 48.33 | 49.17 | 16.67 | 34.44 |
| Gemini-2.5-Flash | 35.04 | 42.50 | 8.57 | 10.00 | 26.67 | 49.17 | 41.67 | 14.17 | 28.89 |
| Qwen3-Coder-30B-A3B-Instruct | 23.93 | 11.67 | 6.67 | 6.67 | 20.00 | 35.00 | 31.67 | 25.00 | 6.67 |
| DOCKSMITH | **51.28** | **51.67** | **19.05** | 15.56 | 20.00 | **63.33** | **57.50** | **30.00** | 41.11 |

bench Verified (SWE.V) (Jimenez et al., 2023), consisting of Python issue-resolution tasks with verified correctness; (2) SWE-bench Multilingual (SWE.M) (Zan et al., 2025), extending issue resolution to multiple languages; and (3) Terminal-Bench 2.0 (Terminal) (Merrill et al., 2026), a command-line-centric benchmark that stresses long-horizon tool use and iterative execution.

**Metrics.** We report two primary outcome metrics for the Multi-Docker-Eval benchmark: *Fail-to-Pass (F2P)*, which measures the proportion of tasks where the configured environment successfully transitions tests from a failing to a passing state, and *Commit Rate*, which quantifies the fraction of instances where the agent submits a solution for evaluation, reflecting model confidence rather than ground-truth correctness. In addition, we report several process metrics, including average input tokens, average output tokens, and average Docker image count, to characterize efficiency and resource usage during environment construction. For SWE.V, SWE.M, and Terminal, we follow the standard evaluation protocols of the respective benchmarks and report absolute performance scores.

**Training Details.** All training experiments are initialized from Qwen3-Coder-30B-A3B-Instruct (Team, 2025) and trained under identical optimization settings unless otherwise specified. We use a global batch size of 32, a learning rate of $1 \times 10^{-5}$, and train all models for two epochs. For models trained on mixed trajectories, we set the maximum sequence length to 64K tokens to accommodate long SWE-solving trajectories. In contrast, DOCKSMITH, which is trained exclusively on Docker-building trajectories, uses a shorter maximum sequence length of 32K tokens.

### 3.2. The Performance of Docker Building

Table 1 shows that DOCKSMITH achieves the best overall performance based on agent framework SWE-Factory, reaching 39.72% F2P and a 58.28% Commit Rate, outperforming all evaluated open-source and closed-source baselines. This result exceeds the previously reported upper bound of 37.7% F2P on Multi-Docker-Eval. The improvement of +20.3 absolute F2P points over the base model suggests that the gains primarily arise from the training procedure rather than increased model scale.

From an efficiency perspective, DOCKSMITH maintains competitive process metrics, with moderate average input/output token usage and a controlled Docker image size, despite achieving substantially higher success rates. Compared to strong baselines such as DeepSeek-v3.1 (Liu et al., 2024a) and Kimi-K2-0905 (Team et al., 2025), DOCKSMITH trades slightly increased resource consumption for a clear gain in configuration reliability, suggesting a favorable effectiveness–efficiency balance.

The Language-level results in Table 2 further show that DOCKSMITH delivers consistent improvements across most ecosystems, with particularly strong gains in Python, JavaScript, Go, and Ruby—languages characterized by standardized dependency management and testing workflows. These patterns align with the benchmark's analysis that configuration success is strongly influenced by ecosystem regularity, while low-level systems languages (e.g., C/C++, Java, Rust) remain bottlenecked by compiler toolchains and system dependencies. Overall, the results demonstrate that DOCKSMITH substantially advances the state of the art on Multi-Docker-Eval, validating its effectiveness as a more reliable "shovel" for large-scale, automated SWE pipelines.

*Table 3.* Ablation of loop-detection and cross-task memory components. All variants use the SWE-Factory framework. F2P (%) on Multi-Docker-Eval.

| Variant | GPT-5-Mini | Qwen3-Coder | DOCKSMITH |
|---|---|---|---|
| SWE-Factory | 33.23 | 17.07 | 36.23 |
| + Loop-detection | 33.98 | 17.07 | 36.53 |
| + Memory pool | 34.13 | 18.57 | 37.73 |
| + Loop & Memory | **34.13** | **19.46** | **39.72** |

### 3.3. Contribution of Loop Detection and Cross-Task Memory

To isolate the individual contributions of the loop-detection controller and the cross-task memory pool to DOCKSMITH's environment-building performance, we conduct a controlled ablation study across three model families: **(i)** GPT-5-Mini (strong closed-source baseline), **(ii)** Qwen3-Coder-30B-A3B-Instruct (our un-finetuned backbone), and **(iii)** DOCKSMITH (our fine-tuned model). All evaluations are conducted on Multi-Docker-Eval using the SWE-Factory framework.

Table 3 reports the F2P results. For DOCKSMITH, the individual components provide moderate gains—Loop-detection alone contributes +0.30 and Memory alone contributes +1.50—but their combination yields a strong synergistic effect of +3.49 absolute F2P over the barebone framework. This synergy arises because loop-detection actively breaks stagnant repair cycles, forcing the agent to explore alternative strategies, which in turn maximizes the utilization of verified solutions retrieved from the cross-task memory pool.

On the un-finetuned Qwen3-Coder backbone, adding loop-detection alone yields no improvement (17.07%), as simply breaking a loop often causes an untrained model to make a novel mistake. However, when paired with the memory pool, this control intervention successfully guides the model to adopt verified trajectories, boosting performance to 19.46%. Even for the strong closed-source GPT-5-Mini, both components provide stable, complementary gains (33.23% → 34.13%), confirming that the architecture generalizes across capability tiers and model families.

These results validate that loop-detection and cross-task memory are both *necessary* and *synergistic* components: loop-detection improves computational efficiency by preventing unproductive repair loops, while cross-task memory enables zero-shot reuse of verified solutions and drives compounding success gains as the memory pool grows.

### 3.4. Docker-Building Trajectories For Training

**Does Docker-Building Improve Agentic Ability?** We conduct a controlled ablation study to examine the effect of jointly training on SWE-solving trajectories and Docker-building trajectories under different token-level mixing ratios. Figure 2 reports performance changes measured as deltas relative to a SWE-only baseline trained exclusively on SWE-solving trajectories. The exact numerical values are provided in the Appendix A.1. The SWE-solving trajectories are drawn from the Nex Agent-SFT dataset (Cai et al., 2025), and their total token budget is held constant across all settings to isolate the impact of incorporating Docker-building trajectories. We vary the number of tokens allocated to Docker-building trajectories to systematically assess how different mixing ratios influence downstream performance. We evaluate the resulting models on SWE.V, SWE.M, and Terminal. Across all three benchmarks, adding Docker-building trajectories can improve downstream agentic performance, but the benefit depends on the token-level mixing ratio between SWE-solving and Docker-building data. Here, ratios are written as SWE : Docker: for example, 1:0.25 means adding 0.25 Docker-building tokens per SWE token while keeping the SWE token budget fixed.

On SWE.V and SWE.M, moderate mixing ratios yield the most stable gains: performance improves monotonically from low ratios (1:0.125, 1:0.25) to more balanced settings, peaking around a 1:1 ratio with gains of +2.25 and +2.09, respectively. These results suggest that Docker-building trajectories enhance the model's ability to reason about dependencies, execution order, and failure recovery—capabilities that transfer directly to SWE-style bug fixing and validation tasks. In contrast, overly skewed mixtures (e.g., 1:2) slightly degrade SWE performance, indicating that excessive emphasis on environment-centric behaviors can dilute task-specific reasoning signals.

Beyond SWE benchmarks, results on Terminal-Bench 2.0 reveal a stronger and more asymmetric effect. Terminal performance improves substantially under Docker-augmented training, with the largest gain (+3.37) observed at a 1:0.5 ratio, followed by sustained improvements at 1:0.25 and 1:2. This highlights that Docker-building trajectories function as a powerful form of agentic skill training, strengthening the model's competence in command execution, tool invocation, and long-horizon interaction.

Overall, these results indicate that Docker-building trajectories provide *complementary supervision* and support environment construction as a transferable training signal beyond preprocessing, bridging static code reasoning with dynamic, tool-driven execution.

**Does SWE Data Improve Docker-Building Ability?** We examine whether SWE-solving trajectories help Docker-building by comparing two controlled settings: training on Docker-building trajectories only, and joint training that adds SWE-solving trajectories at a fixed ratio of 0.5 SWE tokens per Docker token. Table 4 shows that Docker-only

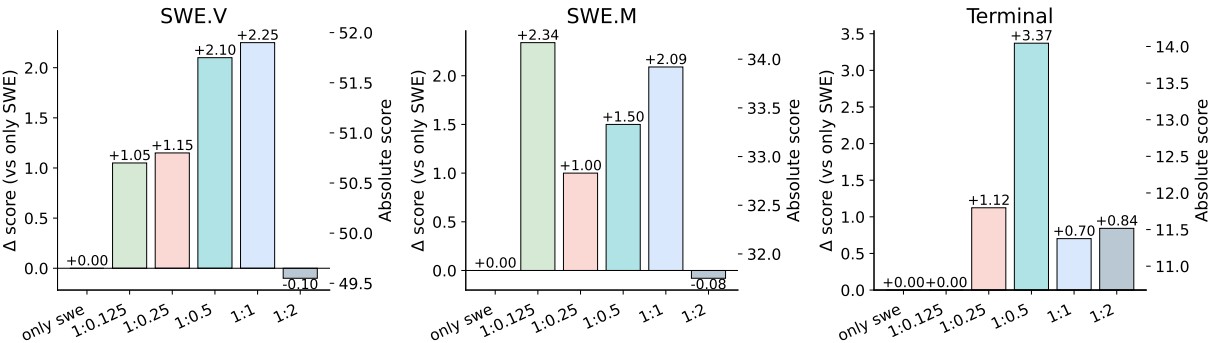

*Figure 2.* Impact of mixing Docker-building trajectories during training. Mixing ratios are defined at the token level and written as SWE : Docker: for example, 1:0.25 means adding 0.25 Docker-building tokens per SWE token, while keeping the SWE token budget fixed. Bars report Δ scores relative to an SWE-only baseline on SWE.V, SWE.M, and Terminal-Bench 2.0.

*Table 4.* Impact of adding SWE-solving trajectories to Docker-building training.

| Data | SWE.V | SWE.M | Terminal | MDE |
|---|---|---|---|---|
| Docker | 33.50 | 18.25 | 7.16 | 34.43 |
| Docker + 0.5× SWE | 47.45 | 31.00 | 10.11 | 35.63 |

*Table 5.* Multi-Docker-Eval F2P (%) under acceptance shaping (AS) and complexity-based curriculum sampling (CC), mean ± std. over 10 runs. All variants use the same training budget.

| Variant | Easy | Hard | Avg |
|---|---|---|---|
| Baseline | $45.22 \pm 1.42$ | $29.18 \pm 0.54$ | $32.40 \pm 0.61$ |
| + AS | $46.42 \pm 1.79$ | $30.86 \pm 0.62$ | $33.98 \pm 0.65$ |
| + CC | $45.37 \pm 1.04$ | $30.30 \pm 0.94$ | $33.32 \pm 0.89$ |
| + AS + CC | $\mathbf{47.31 \pm 1.58}$ | $\mathbf{31.87 \pm 0.84}$ | $\mathbf{34.97 \pm 0.88}$ |

training yields reasonable MDE performance but substantially weaker results on SWE benchmarks. Adding SWE-solving trajectories improves all benchmarks, including consistent gains on MDE and marked improvements on SWE.V, SWE.M, and Terminal. This suggests that SWE trajectories provide complementary task-level signals that better align Docker-building with downstream execution and validation, rather than optimizing environment setup in isolation.

### 3.5. Effect of Trajectory Filtering and Curriculum Sampling

To isolate the effect of *trajectory selection* from training compute, we run a strictly controlled ablation in which all variants are trained under an identical optimization budget, with the same total number of training tokens and epochs. As a result, any performance differences can be attributed to differences in how trajectories are filtered and sampled, rather than to additional data or compute.

We evaluate four training variants: (i) *Baseline*, which trains

on a randomly sampled subset of resolved Docker-building trajectories without additional filtering; (ii) *Baseline + Acceptance Shaping* (*AS*), which applies the acceptance shaping strategy in Section 2.3.1 to remove excessively long or redundant rollouts, then resamples to match the same token budget; (iii) *Baseline + Complexity Curriculum* (*CC*), which applies the complexity-based curriculum sampling scheme in Section 2.3.1 to enforce a balanced difficulty distribution under the same token constraint; and (iv) *Baseline + AS + CC*, which first denoises trajectories via acceptance shaping and then constructs a curriculum-balanced set with a 1:2:2 Easy/Medium/Hard ratio.

As shown in Table 5, applying *Acceptance Shaping* consistently improves F2P over the baseline across difficulty levels. This suggests that excessively long and repetitive rollouts introduce noise under a fixed token budget, and that pruning such trajectories leads to more effective gradient updates and more stable learning. *Complexity Curriculum* also yields clear gains, with improvements concentrated on Hard instances. This indicates that naive random sampling tends to over-represent trivial builds, whereas enforcing a difficulty-aware sampling strategy increases exposure to challenging environment configurations that are critical for generalization. Notably, combining the two techniques (AS + CC) produces the strongest overall performance. This result highlights their complementary roles: acceptance shaping improves the *quality* of individual trajectories by removing redundancy, while curriculum sampling improves the *coverage* of the training distribution by balancing difficulty. Together, they yield consistently better Docker-building performance and more effective training. Sensitivity analysis in Appendix A.2 further confirms that single-dimension proxies collapse the curriculum and that Z-score normalization achieves the global optimum.

*Table 6.* Docker-building error analysis on Multi-Docker-Eval. Error counts are aggregated across all trajectories, and ↓ indicates lower is better. DOCKSMITH achieves consistent reductions across most categories, with a slight increase in Context Retrieval errors due to more aggressive information gathering.

| Metric | Baseline | DOCKSMITH | Δ (%) |
|---|---|---|---|
| *Aggregated Statistics (across all trajectories)↓* | | | |
| Total Errors | 3,757 | **2,161** | −42.5% |
| Avg Errors / Traj | 11.81 | **6.67** | −43.5% |
| Critical Errors | 2,454 | **1,468** | −40.2% |
| Minor Errors | 1,303 | **693** | −46.8% |
| *Error Distribution by Pipeline Stage↓* | | | |
| Dockerfile Generation | 1,434 | **765** | −46.7% |
| Eval Script Handling | 936 | **536** | −42.7% |
| Test Analysis | 1,102 | **544** | −50.6% |
| Context Retrieval | **285** | 316 | +10.9% |

## 3.6. Analysis

### 3.6.1. ERROR ANALYSIS FOR DOCKER BUILDING

To quantify how DOCKSMITH alters failure behavior during Docker building, we conduct a fine-grained error analysis on Multi-Docker-Eval trajectories. GPT-5.1 is used as an automated annotator to label error events across four stages—Context Retrieval, Dockerfile Generation, Eval Script Handling, and Test Analysis—and to assign each event an error code and severity. The full annotation rubric is provided in Appendix C.2.

Table 6 summarizes the aggregated results. DOCKSMITH reduces the total number of errors from 3,757 to 2,161 (−42.5%) and decreases critical errors by 40.2%, with the mean errors per run dropping from 11.81 to 6.67. Improvements are concentrated in stages that directly affect build correctness: Dockerfile-generation errors decrease by 46.7%, Eval-Script errors by 42.7%, and Analysis-stage errors by 50.6%, while Context Retrieval shows a modest increase (+10.9%) due to a higher-recall retrieval strategy that does not offset downstream gains.

### 3.6.2. PER-CODE ERROR BREAKDOWN

Figure 3 further decomposes errors by code and highlights the categories with the largest reductions. The strongest improvement is on `eval_patch_error` (−79.1%): DOCK-SMITH applies evaluation patches more reliably and avoids a recurrent baseline failure mode in which files are re-checked out after patching, silently discarding modifications. Diagnostic quality also improves markedly, with `diag_vague_error` reduced by 71.7% as explanations become more specific and actionable, and `diag_loop_error` reduced by 59.8% as the model avoids oscillating between incompatible repair strategies. Finally, `docker_env_error` drops by 48.7%, indicating fewer environment-configuration mistakes such as missing system packages or misconfigured toolchains.

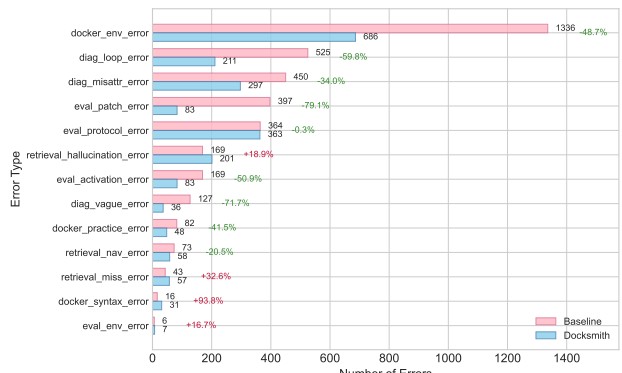

*Figure 3.* Docker-building error counts by error type on Multi-Docker-Eval (aggregated over all trajectories), comparing the baseline and DOCKSMITH. Percent annotations indicate relative changes.

*Table 7.* Error metrics computed at the error-event level. Terminal rate is the fraction of events labeled `Terminal` (lower is better), and resolution rate is the fraction labeled `Resolved`. Overall pools *all* error events across layers and is not derived from the layer-wise rows (e.g., not a simple average).

| Scope | Terminal Rate (%)↓ | | | Resolution Rate (%)↑ | | |
|---|---|---|---|---|---|---|
| | Baseline | DOCKSMITH | Δ | Baseline | DOCKSMITH | Δ |
| *Layer-wise (within-layer rates)* | | | | | | |
| `shell_error` | 3.1 | **2.8** | −0.3 | **60.7** | 58.3 | −2.4 |
| `env_error` | 10.3 | **9.4** | −0.9 | **45.8** | 41.2 | −4.6 |
| `runtime_error` | 6.7 | **5.3** | −1.4 | 46.5 | **49.8** | +3.3 |
| `logic_error` | 22.6 | **15.9** | −6.7 | 44.0 | **52.7** | +8.7 |
| *Overall (pooled across all layers)* | | | | | | |
| Overall | 8.7 | **7.1** | −1.6 | 75.7 | 75.6 | −0.1 |

Across a single Multi-Docker-Eval run, DOCKSMITH performs better on 72.4% of trajectories, and the error distribution shifts toward low-error runs: trajectories with 0–4 errors increase from 49 to 128, while trajectories with 20+ errors decrease from 42 to 8. Overall, the error analysis suggests that DOCKSMITH improves Docker-building reliability through more accurate dependency configuration, convergent diagnostic reasoning, and more robust evaluation-patch handling.

We further analyze how DOCKSMITH changes error propagation and recovery patterns during agentic task execution in Appendix A.3. Key findings include ∼ 6% higher within-layer persistence at the environment and runtime layers, a reduced overall terminal error rate (8.7% → 7.1%), and a shift toward principled rather than heuristic repair strategies (27.1% → 33.0%).

### 3.6.3. TERMINAL AND RESOLUTION OUTCOMES

To understand how DOCKSMITH improves agentic task completion, we annotate error events in execution traces across SWE-bench Verified, SWE-bench Multilin-

gual, and Terminal-Bench 2.0 with a four-layer taxonomy (`shell_error`, `env_error`, `runtime_error`, `logic_error`); full annotation guidelines are in Appendix C.1.

Table 7 reports terminal and resolution rates. DOCKSMITH reduces terminal rates across all layers, with the largest gain on `logic_error` (−6.7 points). For resolution, DOCKSMITH improves `runtime_error` (+3.3%) and `logic_error` (+8.7%), indicating stronger transferable debugging skills, while the baseline retains an edge on `shell_error` (−2.4%) and `env_error` (−4.6%). Overall, the terminal rate drops from 8.7% to 7.1%, confirming net positive robustness gains. A deeper analysis of error propagation and response quality is provided in Appendix A.3.

## 4. Related Works

**Coding LLMs and Agents.** The foundation of autonomous software engineering lies in powerful LLMs, including proprietary models (Anthropic, 2025b; Google, 2025; OpenAI, 2025) and open-source alternatives (Liu et al., 2025; Zeng et al., 2025; Team et al., 2025; Grattafiori et al., 2024; MiniMax, 2025; Ma et al., 2025a; 2024), which achieve high performance through large-scale code pretraining and extended context windows. To enable these models to navigate repository-scale environments, they are frequently augmented with diverse tools. This architectural foundation supports the development of agentic frameworks—such as Agentless (Xia et al., 2024), SWE-agent (Yang et al., 2024), and OpenHands (Team, 2024)—which serve as scaffolds to streamline interactions with environments. Consequently, evaluation paradigms have shifted from isolated function synthesis (Jain et al., 2024; Liu et al., 2023; 2024c; Chen et al., 2021) toward realistic GitHub issue resolution. Benchmarks such as SWE-bench (Jimenez et al., 2023) now serve as the industry standard, alongside newer variants focusing on multilingual tasks (Zan et al., 2025) and feature augmentation (Deng et al., 2025).

**Automatic Environment Setup.** Reliable environment construction is a critical prerequisite for autonomous software maintenance, yet it remains a major bottleneck in the development pipeline (Kovrigin et al., 2025; Hu et al., 2025; Peng et al., 2024). Early LLM-based systems such as ExecutionAgent (Bouzenia & Pradel, 2025) attempted to automate setup across multiple languages but often failed to meet the specialized requirements needed for precise issue reproduction, while more targeted tools like SetupAgent (Vergopoulos et al., 2025) rely on manually defined log parsers and remain closed-source. To address these gaps, SWE-Factory (Guo et al., 2025) proposes a fully automated and multi-agent pipeline that integrates binary file recovery with SWE-Builder, replacing manual verification

with standardized, exit-code-based parsing to enable scalable generation of high-quality training data. In parallel, evaluation benchmarks have evolved: EnvBench (Eliseeva et al., 2025) introduced early metrics, and Multi-Docker-Eval (Fu et al., 2025) now provides a more rigorous benchmark spanning 39 repositories and 9 languages, showing that environment setup remains the primary hurdle for modern open-source models, with success rates often below 40%. Despite steady progress, prior work largely treats environment construction as a supporting step for evaluation and debugging; however, building executable environments inherently involves multi-step reasoning, system configuration, and iterative failure recovery, mirroring broader software engineering challenges and motivating its treatment as a first-class, execution-grounded task in autonomous code agent design.

## 5. Conclusion

Reliable Docker-based environment construction is a major bottleneck for scaling execution-grounded training and evaluation of software engineering agents. We argue that reliable environment construction is not merely a preprocessing step, but a core agentic capability that shapes execution-grounded learning. We introduce DOCKSMITH, a specialized Docker-building agent trained on large-scale, execution-grounded environment-construction trajectories. By modeling environment setup as a long-horizon, verifiable task involving dependency reasoning, tool use, and failure recovery, DOCKSMITH substantially improves Docker build reliability on Multi-Docker-Eval, achieving open-source state-of-the-art performance at a fixed 30B-A3B scale. Beyond environment setup, Docker-building supervision transfers to downstream agentic software engineering: joint training improves performance on SWE-bench Verified, SWE-bench Multilingual, and Terminal-Bench, with reduced error propagation and more stable execution-driven recovery. These results position environment construction as a transferable training signal for scaling autonomous software engineering agents.

## Impact Statement

This paper advances machine learning for autonomous software engineering by improving the reliability of Docker-based environment construction and the scalability of execution-grounded training data for code agents. More dependable environment setup can support more reproducible evaluation and reduce friction in building and testing real-world repositories, potentially benefiting developer productivity and software quality.

At the same time, stronger automation does not remove the need for human oversight and secure development practices. As with other coding-assistant technologies, these

capabilities could be misused to accelerate the development of harmful software; we therefore emphasize responsible release practices and careful downstream deployment.

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

*Table 8.* Effect of different mixing ratios between SWE-solving and Docker-building trajectories during training. Mixing ratios are defined at the token level and written as SWE : Docker.

| Ratio | SWE.V | SWE.M | Terminal |
|---|---|---|---|
| Only SWE | 49.65 | 31.83 | 10.67 |
| 1:0.125 | 50.70 | 34.17 | 10.67 |
| 1:0.25 | 50.80 | 32.83 | 11.80 |
| 1:0.5 | 51.75 | 33.33 | 14.04 |
| 1:1 | 51.90 | 33.92 | 11.38 |
| 1:2 | 49.55 | 31.75 | 11.52 |

*Table 9.* Sensitivity analysis of the complexity metric. All variants use the same training budget.

| Proxy | Easy F2P | Hard F2P | Avg F2P |
|---|---|---|---|
| Length-only | 41.80 | 28.10 | 30.80 |
| RUN-only | 41.80 | 28.80 | 31.40 |
| Pkg-only | 38.80 | 29.20 | 31.10 |
| Z-score (Ours) | **45.30** | **30.10** | **33.10** |

# A. Additional Experiment Results

## A.1. Exact Results of Mixed Training

The exact results are shown in Table 8. The mixing ratio is measured by the token-level proportion between SWE-solving and Docker-building trajectories. Overall, moderate mixing ratios (e.g., 1:0.5–1:1) yield consistent improvements across all three benchmarks, while overly large Docker-building ratios lead to diminishing or unstable gains.

## A.2. Sensitivity Analysis of the Complexity Metric

To assess the robustness of our curriculum design, we evaluate alternative single-dimension proxies against the full Z-score-based complexity score under an identical training budget. As shown in Table 9, relying on a single feature collapses the curriculum: *Length-only* over-samples verbose but trivial configurations (lowest Hard F2P: 28.10%), while *Pkg-only* causes over-engineering on simple setups (dropping Easy F2P to 38.80%). The balanced Z-score normalization correctly captures structure, command chains, and dependency complexity simultaneously, achieving the global optimum across both Easy and Hard splits.

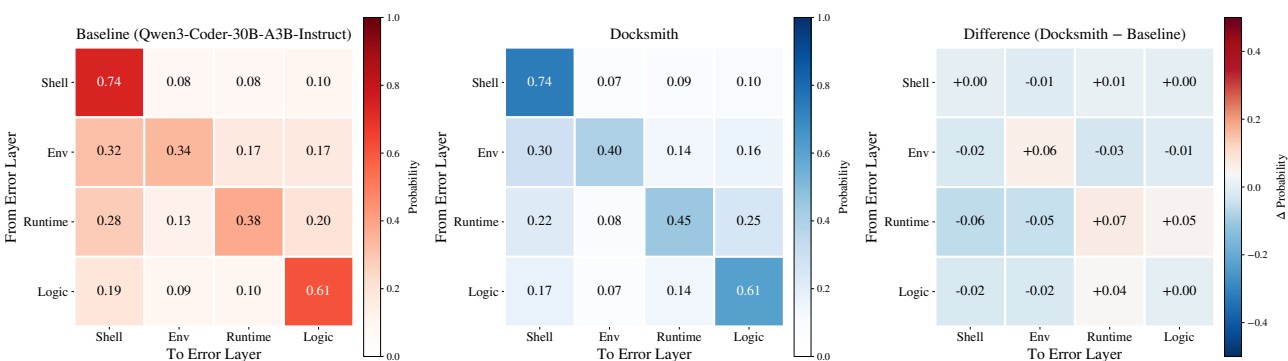

*Figure 4.* Error propagation probability matrices aggregated across SWE-bench Verified, SWE-bench Multilingual, and Terminal-Bench 2.0. Each cell shows the probability that an error at the row layer transitions to an error at the column layer in the next step. **Left**: Baseline. **Center**: DOCKSMITH. **Right**: Difference (Δ), where blue indicates lower propagation and red indicates higher propagation.

*Table 10.* Response quality analysis at the error-event level: intent match quality and repair rationale distribution (%).

| Metric | Baseline | DOCKSMITH | Δ |
|---|---|---|---|
| *Intent Match Quality* | | | |
| High (Precise) | 34.6 | **40.4** | +5.8 |
| Medium (Tentative) | **36.5** | 36.4 | −0.1 |
| Low (Misguided) | 28.8 | **23.1** | −5.7 |
| *Repair Rationale* | | | |
| Principled (System) | 27.1 | **33.0** | +5.9 |
| Heuristic (Log-driven) | 66.3 | **61.8** | −4.5 |
| Blind (Random) | 6.5 | **5.1** | −1.4 |

## A.3. Error Propagation and Recovery in Agentic Tasks

To complement the terminal and resolution analysis in the main text, we provide a detailed breakdown of error propagation patterns and response quality across SWE-bench Verified, SWE-bench Multilingual, and Terminal-Bench 2.0. We annotate execution traces with a hierarchical taxonomy covering four software-stack layers (`shell_error`, `env_error`, `runtime_error`, `logic_error`), together with intent match quality and repair rationale annotations; full annotation guidelines are in Appendix C.1.

**Cross-layer propagation vs. within-layer persistence.** Figure 4 shows error transition matrices for DOCKSMITH and the baseline Qwen3-Coder-30B-A3B-Instruct. Each entry $(i, j)$ gives the probability that an error at layer $i$ is followed by an error at layer $j$ in the next interaction step. Higher diagonal values indicate stronger *within-layer persistence*: the agent remains focused on the same class of failures rather than prematurely shifting context. DOCKSMITH shows consistently stronger persistence at the environment and runtime layers: $P(\texttt{env} \to \texttt{env})$ increases from 0.34 to 0.40 (+6%), and $P(\texttt{runtime} \to \texttt{runtime})$ increases from 0.38 to 0.45 (+7%). This indicates more layer-consistent diagnosis, with the agent iteratively refining dependency and configuration fixes instead of switching to unrelated debugging strategies.

**Response quality and repair rationale.** Table 10 analyzes how the agent responds to observed failures. DOCKSMITH increases the fraction of high-quality, root-cause-aligned responses from 34.6% to 40.4% (+5.8%), while reducing low-quality (misguided) responses from 28.8% to 23.1% (−5.7%). Moreover, DOCKSMITH shifts the repair rationale toward principled system reasoning (27.1%→33.0%) and away from purely heuristic log-driven trial-and-error (66.3%→61.8%). Together with the error propagation analysis and the terminal/resolution statistics in the main text, these results suggest that Docker-building trajectories promote more mechanism-grounded debugging behaviors that remain focused on the relevant failure layer, reducing spurious cross-layer transitions.

DOCKSMITH improves end-to-end robustness in two ways: (i) it exhibits stronger within-layer persistence for environment and runtime failures, and (ii) it increases the fraction of high-quality, root-cause-aligned responses, with the largest gains on runtime and logic debugging. The mixed effects at the shell and environment layers suggest a trade-off between trajectory-level reasoning and low-level command precision, indicating avenues for future work on fine-grained command execution without sacrificing higher-level diagnostic reasoning.

## A.4. EnvBench Evaluation

To further validate the generalization of DOCKSMITH beyond PR/issue-level environment construction to repository-level setups, we evaluate on EnvBench (Eliseeva et al., 2025), a benchmark designed to assess automated environment configuration at the repository scale. We compare DOCKSMITH against both our unfinetuned backbone (Qwen3-Coder-30B-A3B-Instruct) and the strongest baseline reported in the original EnvBench paper (GPT-4o with Bash Agent). Both models are evaluated under the SWE-Factory framework.

*Table 11.* EnvBench evaluation results. `pass@1` measures the fraction of repositories where the built environment passes the verification tests; `avgErrs` reports the mean number of build errors (lower is better), with the number of evaluated repositories in parentheses for JVM.

| Model | JVM pass@1 ↑ | JVM avgErrs ↓ | Python pass@1 ↑ | Python avgErrs ↓ |
|---|---|---|---|---|
| EnvBench SOTA (GPT-4o Bash) | 29.47% | 26.84 (216) | 6.69% | 52.00 |
| Qwen3-Coder-30B-A3B-Instruct | 10.53% (70/665) | 2.63 (27) | 10.94% | 14.39 |
| DOCKSMITH (Ours) | **73.08%** (486/665) | 6.04 (166) | **24.92%** | **11.09** |

*Table 12.* Trajectory-level comparison on a representative Multi-Docker-Eval case (`JEG2/highline` Issue #222), contrasting the base model (Qwen3-Coder-30B-A3B-Instruct) and DOCKSMITH.

| Metric | Baseline | DOCKSMITH | Change |
|---|---|---|---|
| Total steps | 50 | 5 | $-90.0\%$ |
| Total errors | 37 | 2 | $-94.6\%$ |
| Critical errors | 14 | 2 | $-85.7\%$ |
| Diagnostic loops | 3 | 0 | $-100\%$ |
| Dockerfile rollback events | 2 | 0 | $-100\%$ |

As shown in Table 11, DOCKSMITH achieves 73.08% pass@1 on JVM and 24.92% on Python, substantially outperforming both the base model and the original EnvBench SOTA (GPT-4o Bash Agent). This indicates that the skills learned from PR/issue-level trajectories effectively generalize to repository-level environment setups.

It is important to note that the lower JVM avgErrs of the base model (2.63) is an artifact of selection bias—it only successfully produced countable errors on 27 easy Maven repositories. DOCKSMITH, by contrast, evaluates across 166 significantly harder repositories, accumulating more intermediate errors (6.04) while achieving a vastly higher pass@1 (73.08% vs. 10.53%). When both models are evaluated fairly on the exact same overlapping Python subset ($N = 85$), DOCKSMITH is proven to be functionally more accurate, producing strictly fewer errors (11.09 vs. 14.39).

# B. Case Study

## B.1. Case Study: Multi-Docker-Eval Trajectory Analysis

To qualitatively illustrate how Docker-building supervision alters agentic behavior, we present a representative case study from the Multi-Docker-Eval benchmark. We compare the base model, **Qwen3-Coder-30B-A3B-Instruct**, with DOCKSMITH on a single repository-level task, analyzing their full execution trajectories under identical evaluation protocols.

The task is drawn from a Ruby repository (`JEG2/highline`), corresponding to Issue #222. Successfully completing this instance requires constructing a valid Docker environment, resolving native dependency compilation, and executing the project's test suite. In particular, the `rugged` gem introduces non-trivial system-level dependencies, including `cmake`, `pkg-config`, and OpenSSL development libraries, making this a representative environment-construction challenge.

**Summary of Outcomes.** Table 12 summarizes the trajectory-level statistics for this case. Compared to the base model, DOCKSMITH drastically shortens the rollout and reduces error accumulation: it completes the task in 5 steps with only 2 errors, whereas the baseline reaches the step limit (50 steps) while triggering 37 errors. In addition, DOCKSMITH eliminates all diagnostic loops and Dockerfile rollback events observed in the baseline, indicating substantially more stable iteration behavior.

The baseline trajectory exhibits several recurring failure patterns.

**(1) Dockerfile rollback and state inconsistency.** The model repeatedly generates incomplete Dockerfiles, temporarily adds missing dependencies, and then reverts to earlier versions, reintroducing the same errors. This behavior indicates a lack of persistent state tracking across iterations.

**(2) Incremental and fragmented dependency diagnosis.** Rather than reasoning about native dependencies holistically, the baseline discovers missing components one at a time (e.g., `cmake` → `pkg-config` → OpenSSL), resulting in trial-and-error exploration and unnecessary iterations.

**(3) Diagnostic oscillation.** The analysis agent alternates between incompatible testing options without validating their semantics, entering repeated loops between invalid command configurations.

**(4) Evaluation protocol violations.** Despite explicit requirements, the baseline repeatedly omits environment activation steps in the evaluation script, leading to a large number of protocol-level errors.

In contrast, DOCKSMITH exhibits substantially more stable and convergent behavior.

**(1) One-shot dependency resolution.** DOCKSMITH identifies the complete set of required native dependencies for `rugged`

in a single pass and produces a correct Dockerfile without rollback.

**(2) Loop avoidance and decision stability.** The trajectory contains no oscillatory diagnosis patterns or repeated invalid strategy switches, indicating more disciplined repair behavior.

**(3) Improved protocol adherence.** Protocol violations are reduced from 20 occurrences to a single instance, reflecting a stronger alignment with execution constraints.

This case study highlights three qualitative shifts induced by Docker-building supervision: (i) improved state consistency across iterations, (ii) more systematic dependency reasoning, and (iii) reduced diagnostic oscillation. Together, these changes lead to faster convergence and substantially cleaner execution traces, even on dependency-heavy repositories.

**Trajectory Visualization.** For completeness, we include the raw execution trajectories for both models below.

*Listing 1.* Baseline trajectory for `JEG2/highline` Issue #222 (Qwen3-Coder-30B-A3B-Instruct).

```
Trajectory (abridged; key steps only)
Repo: JEG2/highline
Commit: 413ad89036ab535e53642c66333e9622323fdf32
Source: baselie_applicable_setup_JEG2__highline-222.json
Total steps: 50

[Key steps]
Step 1 | Context Retrieval Agent
  User: Target repository name: JEG2/highline; Commit SHA: 413
      ad89036ab535e53642c66333e9622323fdf32; task: gather setup context
  Asst: Plans to inspect README/Gemfile/Rakefile/gemspec/CI config to infer env + test
      commands.

Step 13 | Write Eval Script Agent
  Assistant (excerpt): writes eval script skeleton with:
    cd /testbed
    git checkout 413ad89036ab535e53642c66333e9622323fdf32
    git apply -v - <<'EOF_114329324912' [CONTENT OF TEST PATCH] EOF_114329324912
    ... (run tests) ...
    rc=$?
    echo "OMNIGRIL_EXIT_CODE=$rc"

Step 41 | Test Analysis Agent (Docker build failure)
  Log (excerpt):
    Installing rugged 0.99.0 with native extensions
    checking for cmake... no
    ERROR: CMake is required to build Rugged.
  Guidance: add 'cmake' to apt deps in Dockerfile.

Step 42 | Test Analysis Agent (Docker build failure after cmake)
  Log (excerpt):
    checking for cmake... yes
    checking for pkg-config... no
    ERROR: pkg-config is required to build Rugged.
  Guidance: add 'pkg-config' to apt deps in Dockerfile.

Step 44 | Test Analysis Agent (Test failure)
  Symptom: failing test TestHighLine#test_readline_mode
    expected pattern '/Prompt:\ \ any\ input\n/' but got 'Prompt: '
  Guidance: environment/readline simulation issue or brittle test; review test logic +
      patch compatibility.

Step 49 | Test Analysis Agent (Eval script failure)
  Symptom: minitest reports invalid option '--exclude-pattern=test_readline_mode'
  Guidance: use minitest-supported exclude flag (e.g., '-e/--exclude'), not '--exclude-
      pattern'.

Step 50 | Test Analysis Agent (Eval script failure)
  Symptom: "File does not exist: /testbed/test_readline_mode"
```

```
Guidance: the exclusion/selection syntax is being interpreted as a file path; fix how
    TESTOPTS is passed.
```

*Listing 2.* DOCKSMITH trajectory for `JEG2/highline` Issue #222.

```
Trajectory (abridged; key steps only)
Repo: JEG2/highline
Commit: 413ad89036ab535e53642c66333e9622323fdf32
Source: it2520_applicable_setup_JEG2__highline-222.json
Total steps: 5

[Key steps]
Step 1 | Context Retrieval Agent
  Assitant: inspects key files/dirs to determine Ruby version, deps, and test commands.

Step 2 | Context Retrieval Agent
  User: repository browsing results (lib/, test/, Gemfile/gemspec/CI)
  Assitant: confirms how to run tests + any special requirements.

Step 3 | Write Dockerfile Agent
  Assitant (excerpt): proposes Dockerfile based on Debian Ruby image:
    FROM --platform=linux/x86_64 ruby:2.7-bullseye
    SHELL ["/bin/bash", "-lc"]
    ... (clone repo, checkout commit, bundle install) ...
    CMD ["/bin/bash"]

Step 4 | Write Eval Script Agent
  Assitant (excerpt): eval script skeleton with:
    cd /testbed
    git checkout 413ad89036ab535e53642c66333e9622323fdf32
    git apply -v - <<'EOF_114329324912' [CONTENT OF TEST PATCH] EOF_114329324912
    ... (run tests) ...
    rc=$?
    echo "OMNIGRIL_EXIT_CODE=$rc"

Step 5 | Test Analysis Agent
  Result: OMNIGRIL_EXIT_CODE=0
  is_finish: true
```

## B.2. Case Study: Environment-Aware Transfer on SWE-bench Verified

We present a representative case study from SWE-bench Verified to qualitatively illustrate how joint training with Docker-building trajectories alters agentic execution behavior. The selected instance, `scikit-learn/scikit-learn#25102`, requires modifying the output processing logic to preserve `pandas.DataFrame` dtypes in the pandas output of transformers. Although the code change itself is localized, success hinges on execution-grounded validation against the benchmark's FAIL_TO_PASS tests.

**Baseline behavior.** The baseline model, trained exclusively on SWE-solving trajectories, fails in all four independent repeated runs (0/4; four reruns under the same evaluation setting). A representative run (repeat_0) successfully executes several local checks (e.g., Python version and selected unit tests), but still fails the official evaluation: none of the two FAIL_TO_PASS tests are fixed (0/2).

**Mixed-model behavior.** In contrast, the model trained with mixed SWE-solving and Docker-building trajectories achieves a 50% success rate over the same four reruns (2/4). In a representative successful run (repeat_0), the mixed model performs an explicit early verification of test tooling (e.g., `pytest --version`) and executes a broader set of feature-selection tests before final evaluation. Crucially, it fixes both FAIL_TO_PASS tests (2/2), resolving the dtype-preservation failures that remain in the baseline run.

**Behavioral differences.** Comparing the two compressed traces highlights three systematic differences. First, the mixed model adopts earlier execution-oriented verification, explicitly checking the test runner before iterating on the

patch. Second, it follows a more comprehensive validation routine by running a wider set of relevant tests under `sklearn/feature_selection/`, increasing the chance of catching dtype-related regressions aligned with the benchmark's target failures. Third, the mixed model's final patch more directly enforces dtype propagation from the input `DataFrame` to the pandas output, matching the evaluation signal of the remaining failures.

This case suggests that Docker-building supervision transfers beyond environment setup by strengthening execution-grounded habits: toolchain verification, systematic test execution, and iterating patches against concrete failure signals. While both models can run unit tests, only the mixed model consistently closes the loop between observed test failures and the patch logic required to satisfy the benchmark's FAIL_TO_PASS criteria (2/2 vs. 0/2 in `repeat_0`).

We report a compressed trace for `scikit-learn/scikit-learn#25102` (SWE-bench Verified), comparing Baseline (unresolved) vs Mixed (resolved). We keep only (i) key code edits, (ii) key commands/tests, and (iii) the final evaluation outcome. SWE-bench Verified determines success by whether all FAIL_TO_PASS tests are fixed; this instance has two target failing tests.

*Listing 3.* Baseline trajectory for `scikit-learn/scikit-learn#25102` (unresolved).

```
Baseline trajectory (unresolved)

Goal (from issue): preserve input DataFrame dtypes for pandas output of transformers (
    dtype information like category/int precision).

Key edits (from patch):
 - Modified: sklearn/utils/_set_output.py
 - Added: test_dtype_preservation_final.py

High-level implementation intent (from patch diff):
 - Introduce dtype-preservation path for pandas output by capturing input DataFrame
     dtypes and applying astype on the output.
 - Add optional preserve_dtypes configuration to set_output(..., transform="pandas")
     and store in _sklearn_output_config.

Key executed commands / tests (selected):
 - python --version (environment sanity check)
 - python -m pytest sklearn/utils/tests/test_set_output.py ... -> exit=0 (passed)
 - python -m pytest sklearn/feature_selection/tests/test_feature_select.py ...-> exit=0
     (passed)
 - python -m pytest sklearn/feature_selection/tests/test_univariate_selection.py ... ->
     exit=4 (path not found; non-critical to final verdict)

Final evaluation outcome (ground truth):
 status: unresolved (0/2 FAIL_TO_PASS fixed)
 FAIL_TO_PASS failures:
   - sklearn/feature_selection/tests/test_base.py::test_output_dataframe
   - sklearn/feature_selection/tests/test_feature_select.py::
       test_dataframe_output_dtypes
 Minimal failure signal (from eval log):
   AssertionError: assert dtype('O') == dtype('float32')
   (summary: 2 failed, 59 passed)
```

*Listing 4.* Mixed-model trajectory for `scikit-learn/scikit-learn#25102` (resolved).

```
Mixed trajectory (resolved)

Goal (same): preserve input DataFrame dtypes for pandas output.

Key edits (from patch):
 - Modified: sklearn/utils/_set_output.py

High-level implementation intent (from patch diff):
 - Propagate/capture original input DataFrame dtypes and apply them to the pandas
     output DataFrame during wrapping.

Key executed commands / tests (selected):
```

```
  - python --version
  - python -m pytest --version (early test tooling verification)
  - python -m pytest sklearn/utils/tests/test_set_output.py ... -> exit=0
  - python -m pytest sklearn/feature_selection/tests/test_feature_select.py ... -> exit
      =0
  - python -m pytest sklearn/feature_selection/tests/ ... -> exit=0
  - python -m pytest sklearn/feature_selection/tests/test_univariate_selection.py ... ->
      exit=4 (path not found; non-critical to final verdict)

Final evaluation outcome (ground truth):
  status: resolved (2/2 FAIL_TO_PASS fixed)
  FAIL_TO_PASS successes:
    - sklearn/feature_selection/tests/test_base.py::test_output_dataframe
    - sklearn/feature_selection/tests/test_feature_select.py::
        test_dataframe_output_dtypes
```

## C. Rubric

### C.1. Error Attribution Rubric

This appendix provides the error attribution rubric used in the error propagation and recovery analysis (Appendix A.3). The rubric defines a hierarchical taxonomy that assigns each error event in an execution trajectory to one of four software-stack layers, together with auxiliary annotations for intent alignment, causality, and resolution status. It serves as the standardized guideline for annotating error events and agent responses across all evaluated benchmarks.

*Listing 5.* Layered error attribution rubric.

```
[Task Objective]
Given the provided trajectory data, identify all errors that occur throughout the
    trajectory, and attribute each error according to the following Layered Taxonomy.

[Hierarchical Rubric]
- OS/Shell Layer (E_shell): Command syntax errors, incorrect arguments, or misuse of OS-
    level tools (e.g., tar failures, bash script logic errors).
- Environment & Dependency Layer (E_env): Missing system libraries, third-party
    dependency version conflicts, misconfigured file paths/permissions (e.g., pip
    failures, FileNotFoundError).
- Language Runtime Layer (E_runtime): Interpreter startup failures, missing dynamic
    libraries, Python ModuleNotFoundError (even when dependencies are already installed).

- Application Logic Layer (E_logic): Code executes normally but the application/business
     logic is incorrect (e.g., AssertionError, test cases failing).

[Analysis Dimensions]
For each error point in the trajectory, extract the following dimensions:
- Intent Verification: Analyze whether the Agent's fix action logically and rigorously
    targets the root cause of the error.
- Causality Trace: If the error is caused by a previous step, annotate the source step.
- Status Mapping: Track whether the error is ultimately resolved, or triggers cascading
    failures.

[Output JSON Format]
1. Constraints on error_trace:
    a. If multiple errors occur, analyze each error separately. error_trace MUST be a
        list containing one entry per distinct error event (do not merge errors).
    b. If a single step contains multiple distinct errors, create multiple error_trace
        items sharing the same step_index.
2. Output a JSON object in the following structure:
{
  "trajectory_summary": {
    "total_steps": "integer",
    "final_result": "Success/Failure",
```

```
    "primary_failure_bottleneck": "E_shell/E_env/E_runtime/E_logic"
  },
  "error_trace": [
    {
      "step_index": "index of the step where the error occurs",
      "layer": "assigned layer",
      "description": "brief description of the error",
      "agent_response": {
        "action_taken": "the specific fix action taken by the Agent",
        "intent_match": "High (precise)/Medium (tentative)/Low (random or misguided)",
        "knowledge_source": "System (principle-based)/Empirical (log-guessing)/Random (
            blind)"
      },
      "status": "Resolved/fixed, Terminal/ends task, Cascaded/triggers downstream cross-
          layer errors",
      "causality": "if cascaded, specify which step induced it"
    }
  ],
  "global_metrics": {
    "error_density": "ratio of error steps to total steps",
    "self_correction_efficiency": "number of successfully resolved errors / total number
        of errors",
    "environment_grounding_score": "score from 0 to 1 measuring the Agent's depth of
        environment understanding"
  }
}

[Trajectory Data]
##Insert your trajectory text here##
```

## C.2. Docker-building Error Annotation Rubric

We provide the full rubric used by the automated annotator (GPT-5.1) to label Docker-building trajectories. The rubric specifies (i) agent-specific error taxonomies, (ii) severity labels, and (iii) the required JSON schema for recording error events. We include only steps that contain an error event; steps without errors are omitted.

*Listing 6.* Docker-building error annotation rubric and output schema.

```
[Task Instructions]

Analyze the trajectory step-by-step. For each step, check the provided [Active Agent]
    label and apply the corresponding logic:
1. Assess Quality: Does the action fail or exhibit bad practices?
2. Apply Taxonomy: Select the specific error code from the list corresponding to the [
    Active Agent].
3. Trace Causality: If an error occurs, determine if it is a direct consequence of a
    previous agent's output (e.g., Analysis Agent failed because Context Agent provided
    wrong info).

[Agent-Specific Error Taxonomies]

Scenario A: [Active Agent] == Context Retrieval Agent
- retrieval_miss_error: Failed to extract existing info (e.g., missed requires in setup.
    py, ignored .python-version).
- retrieval_hallucination_error: Invented non-existent dependencies or configurations.
- retrieval_nav_error: Navigation failure (e.g., looked in wrong directory, failed to
    find file).

Scenario B: [Active Agent] == Write Dockerfile Agent
- docker_syntax_error: Generated invalid Dockerfile instructions (build fails
    immediately).
- docker_env_error: Valid syntax but wrong semantic environment (e.g., missing system
    libs like libpq-dev, wrong base image).
```

```
- docker_practice_error: Violated best practices (e.g., running tests inside RUN
    instruction, pinning mutable tags).

Scenario C: [Active Agent] == Write Eval Script Agent
- eval_activation_error: Failed to activate the virtual environment (venv/conda).
- eval_patch_error: Failed to apply git patches correctly.
- eval_protocol_error: Failed to capture exit codes (OMNIGRIL_EXIT_CODE) or logs
    properly.

Scenario D: [Active Agent] == Test Analysis Agent
- diag_misattr_error: Root Cause Misdiagnosis. Blamed the wrong component (e.g., blamed
    code for an environment issue).
- diag_loop_error: Repeated the exact same failed fix suggestion.
- diag_vague_error: Provided non-actionable feedback (e.g., "Fix the error" without
    saying how).

[Output JSON Format] Output a JSON object containing a list of error events. If a step
    has no error, do not include it.

```
{
  "trajectory_id": "string",
  "error_events": [
    {
      "step_index": integer,
      "active_agent": "Context Retrieval | Dockerfile | Eval Script | Test Analysis",
      "error_code": "E_...",
      "severity": "Critical | Minor",
      "description": "Brief explanation of the failure",
      "causality_trace": {
       "is_cascaded": boolean,
       "root_cause_step": integer_or_null,
       "reasoning": "e.g., Failed here because Step 5 missed the dependency."
      }
    }
  ]
}
```

[Input Data Format]

```
Step 1:
[Active Agent]: Context Retrieval Agent
[Action]: Searching for "requirements.txt" in root directory...
[Output]: File not found.

Step 2:
[Active Agent]: Write Dockerfile Agent
[Action]: ...
```
```

