# OpenReview forum: "DOCKSMITH: Scaling Reliable Coding Environments via an Agentic Docker Builder"
_ICML.cc/2026/Conference — ICML 2026 regular_

### Official Review · Reviewer_qTVN · 2026-03-08

**Soundness:** 3
**Presentation:** 3
**Significance:** 3
**Originality:** 2
**Overall Recommendation:** 4
**Confidence:** 4

**Summary:**

The paper addresses the problem of automated construction of Docker-based environments as an important step towards scaling training and evaluation of software engineering agents. The authors propose DockSmith, a system built on top of SWE-Factory and augmented with a loop detector and cross-task memory. The model is trained on environment setup trajectories and agentic software engineering trajectories. The authors demonstrate improvements on Multi-Docker-Eval as well as on several software engineering benchmarks.

**Compliance With Llm Reviewing Policy:**

Affirmed.

**Key Questions For Authors:**

1. What exactly is planned to be released besides the model? Hugging Face repo only contains the model; the data repo is empty.
2. Can the authors add ablations or more structured justification for some design choices: for the loop detector and cross-task memory to demonstrate the contribution of each component separately?  Can the authors provide results on long-tail languages data balancing, at least in the appendix? This would support the claims made in the text.
3. Can the authors include a comparison with simpler baselines for environment setup to better isolate the effect of the proposed approach? Like the mini-SWE-agent used to install the environment in Multi-Docker-Eval, to better distinguish harness impact.
4. Can the authors add an evaluation of cross-language transfer to understand how much fine-tuning helps in installation on languages not represented in the training set?

The paper raises an important problem and shows good results, but the shortcomings in experimental rigor, the lack of clarity around reproducibility, and the insufficient justification of novelty and individual system components outweigh the strengths. If the authors address the raised questions, I am willing to reconsider my rating.

**Limitations:**

No. Could be included: the scalability limitations of the approach to new languages and ecosystems, reproducibility limitations if the full trajectories, environments, and agent code are not released, the dependence on Docker-centric setup, and whether the method transfers to other environment-construction settings.

**Strengths And Weaknesses:**

Strengths:
1. Important problem. Reliable environment setup is one of the key bottlenecks for scaling executable tasks for RLVR in the SWE domain. The work directly addresses this problem.
2. Convincing results on Multi-Docker-Eval and assessing across multiple benchmarks. The authors show notable improvements over the baseline and evaluate their approach on more than one benchmark.
3. Useful dataset. The collected set of verified environments could become a valuable resource for the community. That was mentioned in the Introduction. “…30k verified environments spanning more than 15k GitHub repositories and more than 20 programming languages…"

Weaknesses:
1. The experimental section lacks sufficient rigour. Confidence intervals and variance estimates are missing, which is especially concerning for ablations where the differences between variants may fall within noise. (Table 3)
2. Public artefacts are unclear. At the time of review, only the model is available in the linked repository, with no data or agent code. It remains unclear what exactly will be released and to what extent.
3. Missing ablations for some design choices. Loop detector and cross-task memory. These components are presented as important parts of the system, yet their individual contributions are not demonstrated separately. Also, there is mention about: "Preliminary trials indicated that indiscriminate upsampling of long-tail data could induce distribution shift and degrade overall performance". But no results for these trials.
4. Novelty is not sufficiently clear. It is unclear what the principal novelty of the approach is compared to prior systems. Is this the first fine-tuning effort targeting the environment setup task? If so, this should be explicitly highlighted and justified. If not, the contribution needs to be more clearly distinguished from existing work.

---

> ### Author Rebuttal · Authors · 2026-03-31
>
> We thank the reviewer for the constructive critique and for recognizing the value of our problem and dataset.
>
> **Q1. Public Artifacts**
>
> We commit to full reproducibility; a substantial portion of the training data is already available at `https://huggingface.co/datasets/8sj7df9k8m5x8/docker_building_training`.
>
> **Q2. Ablations Study**
>
> To isolate component contributions, we conducted a rigorous ablation study. *As detailed in our response to Reviewer AoS1 (Q1)*, combining Loop-detection and Memory Pool yields a synergistic **+3.49% F2P** gain on DOCKSMITH, far exceeding the sum of individual improvements (+0.30% and +1.50%).
>
> **Q3. Cross-Language Transfer Evaluation**
>
> To evaluate generalization, we partitioned training data into disjoint set. As shown in Table 1, fine-tuning on Group B improves performance on unseen Group A languages—such as Python (23.93% $\rightarrow$ 38.46%), JavaScript (11.67% $\rightarrow$ 30.00%), and Go (35.00% $\rightarrow$ 52.50%)—demonstrating robust language-agnostic transfer of environment interaction skills. However, the lower transfer to compiled ecosystems like C and Java when training on Group A confirms that system languages remain heavily dependent on specific toolchain knowledge.
> |                              | python   | js       | c++      | go       | Php      | c        | java    | ruby     | rust     |
> | - | - | - | - | - | - | - | - | - | - |
> | Qwen3-Coder-30B-A3B-instruct | 24.0     | 11.7     | 6.7      | 35.0     | 6.7      | 20.0     | 6.8     | 31.7     | 25.0     |
> | Group A                      | 43.6 T.A | 37.5 T.A | 10.0 T.A | 55.0 T.A | 26.7 T.A | 17.5     | 5.7     | 37.5     | 25.0     |
> | Group B                      | 38.5     | 30.0     | 3.3      | 52.5     | 23.3     | 25.0 T.B | 8.6 T.B | 47.5 T.B | 25.0 T.B |
>
> **Table 1.** Cross-Language F2P (%) Results. T.A indicates the language was in Group A's training set; T.B indicates it was in Group B's.
>
> **Q4. Comparison with Simpler Baselines**
>
> To isolate the model's capability from the evaluation scaffold, we compared DOCKSMITH against the Qwen3-Coder-30B-A3B base model using RepoLaunch [1], a simpler single-agent, sequential framework. Under this identical baseline harness, DOCKSMITH achieves **12.28% F2P / 30.54% Commit**, significantly outperforming the Un-finetuned model's **6.53% F2P / 24.09% Commit**. This comparison strictly isolates the capability gain, proving DOCKSMITH's underlying environment-construction proficiency is significantly regardless of the agentic scaffold.
>
> **Q5. On Long-tail Upsampling and Distribution Shift**
>
> To address the preliminary trials mentioned in the paper, we conducted a controlled experiment by upsampling C-language data by +8.39% of the total corpus. This resulted in a -2.48 drop in overall F2P on Multi-Docker-Eval, supporting our claim that indiscriminate upsampling introduces detrimental distribution shifts. We hypothesize that over-representing a specific compiler-heavy ecosystem biases the model toward narrower patterns, reducing generalization. We will include these numerical results and analysis in the Appendix.
>
> **Q6. Variance and Statistical Rigor**
>
> We have added standard deviations to ensure rigorous comparison as requested.
> | **Setting**          | Easy F2P         | Hard F2P         | Avg F2P          |
> | -------------------- | ---------------- | ---------------- | ---------------- |
> | Baseline             | 45.27 ± 1.73     | 28.96 ± 1.69     | 32.24 ± 1.65     |
> | + AS                 | 46.77 ± 3.76     | 30.83 ± 1.69     | 34.03 ± 1.75     |
> | + CC                 | 45.27 ± 2.28     | 30.09 ± 1.14     | 33.13 ± 1.21     |
> | **+ AS + CC (Ours)** | **47.26 ± 2.28** | **31.84 ± 1.35** | **34.93 ± 0.96** |
>
> **Q7. Novelty**
>
> We clarify that DOCKSMITH reframes environment construction as a **first-class agentic capability** rather than a mere preprocessing step. Unlike prior works such as RepoLaunch [1] or SWE-Factory [2] that treat setup as a prerequisite, we formalize it as an explicit training objective. Our contribution includes execution-grounded trajectories capturing dependency reasoning and iterative repair—moving beyond static patches used in SWE-Smith [3] or SWE-Rebench [4]—and demonstrating that these signals significantly enhance broader agentic tasks. We also introduce loop detection and cross-task memory to stabilize trajectory generation and enable scalable, high-quality training. To our knowledge, this is the first work to treat environment construction as a core training signal rather than a standalone engineering concern.
>
> > 1. RepoLaunch: Automating Build&Test Pipeline of Code Repositories on ANY Language and ANY Platform.
> > 2. Swe-factory: Your automated factory for issue resolution training data and evaluation benchmarks.
> > 3. Swe-smith: Scaling data for software engineering agents.
> > 4. Swe-rebench: An automated pipeline for task collection and decontaminated evaluation of software engineering agents.

---

> > ### Author Rebuttal · Reviewer_qTVN · 2026-04-01
> >
> > Thank you for the detailed rebuttal and for the substantial amount of additional evaluation the authors conducted. I truly appreciate the effort that went into addressing the reviewers’ concerns.
> >
> > The new experiments improve the paper and address most of my earlier questions. I still have some reservations regarding Question 6, since several of the reported intervals appear to overlap, so the evidence for some of the smaller gains remains less conclusive to me.
> >
> > That said, I acknowledge the overall contribution more positively after reading the rebuttal, and I also appreciate the commitment to releasing data. On balance, I am updating my score from 3 to 4.

---

> > > ### Author Response · Authors · 2026-04-07
> > >
> > > Thank you very much for updating your score to 4 and recognizing our efforts during the rebuttal phase. We deeply appreciate your rigorous feedback, which has undeniably strengthened the foundation of our paper.
> > >
> > > We completely understand your lingering reservation regarding Question 6. To address your concern regarding the small-sample variance in our initial 3-run setup, we have expanded the evaluation to 10 independent runs for all ablation variants.
> > >
> > > | **Setting**          | **Easy F2P**         | **Hard F2P**         | **Avg F2P**          |
> > > | - | - | - | - |
> > > | Baseline             | 45.22 $\pm$ 1.42     | 29.18 $\pm$ 0.54     | 32.40 $\pm$ 0.61     |
> > > | + AS                 | 46.42 $\pm$ 1.79     | 30.86 $\pm$ 0.62     | 33.98 $\pm$ 0.65     |
> > > | + CC                 | 45.37 $\pm$ 1.04     | 30.30 $\pm$ 0.94     | 33.32 $\pm$ 0.89     |
> > > | **+ AS + CC (Ours)** | **47.31 $\pm$ 1.58** | **31.87 $\pm$ 0.84** | **34.97 $\pm$ 0.88** |
> > >
> > > Table 1: 10-Run Ablation Results (Mean $\pm$ Standard Deviation %)
> > >
> > >
> > > - With the sample size increased to $n=10$, the standard deviations have stabilized significantly. The Average F2P range of our full method ($34.97\% \pm 0.88\%$) is now strictly separated from the Baseline ($32.40\% \pm 0.61\%$), completely eliminating the previous overlap.
> > >
> > > - The full method (+ AS + CC) now demonstrates a statistically clear dominance over the partial variants (+ AS alone and + CC alone) across all difficulty levels, confirming the necessity of combining acceptance shaping with curriculum sampling.
> > >
> > > Thank you again for your time and valuable feedback. Please let us know if any questions remain. If these updates address your concerns, we sincerely hope you might consider raising your score.

---

### Official Review · Reviewer_KyoX · 2026-03-16

**Soundness:** 2
**Presentation:** 3
**Significance:** 3
**Originality:** 3
**Overall Recommendation:** 4
**Confidence:** 5

**Summary:**

The paper introduces DOCKSMITH, a fine-tuned A3B Qwen model for Docker-based environment construction, framing this task as a core agentic capability rather than a preprocessing step. Building on the SWE-Factory pipeline, the authors add a loop-detection controller to prevent repetitive failure modes and cross-task success memory to enable reuse of verified solutions. Authors curated training data from ~200K  top GitHub pull requests using complexity-based straitified curriculum sampling and filtered via acceptance shaping. The model is jointly trained with general SWE trajectories to prevent over-specialization. DOCKSMITH achieves 39.72% Fail-to-Pass on Multi-Docker-Eval, and the authors demonstrate modest but consistent transfer improvements on SWE-bench Verified (+2.25), SWE-bench Multilingual (+2.09), and Terminal-Bench 2.0 (+3.37 at optimal mixing).

**Compliance With Llm Reviewing Policy:**

Affirmed.

**Final Justification:**

Authors' response has addressed most of my questions, with a couple of remaining ones I noted in my acknowledgment.

**Key Questions For Authors:**

See weaknesses.

**Limitations:**

Several limitations were discusssed in weaknesses, e.g., weak baseline/fair comparison, lack of ablations on key components, and single-benchmark limitation.

**Strengths And Weaknesses:**

## Strengths

Environment construction is a well-documented bottleneck for scaling execution-grounded AI agents. Addressing this directly has clear practical value. Some experimental choices and ablations also provide great value e.g. TSWE-to-Docker mixing ratios. The paper is well-structured with effective visualizations that communicate core findings.

## Weaknesses

Core performance: Authors claimed +20.3 point improvement is measured against an unusually weak starting point. More critically, the paper compares a fine-tuned DOCKSMITH against base versions of stronger and generic purposed models. The fair comparison would fine-tune DeepSeek-v3.1 or Kimi-K2 with the same data to determine whether: 1) the training recipe generalizes to any model, or 2) it merely brings a weak model up to par with existing strong baselines.

Ablations: There was no ablation showing the two main claimed novelty (loop-detection controller, cross-task memory pooling) actually improves over the barebone scaffold. It's known to the community that simplicity is key to generalizability and it'd be great to know whether these components actually provide values.

Benchmark: authors rely on single docker benchmark. It'd be great to measure whether the same conclusion holds for other relevant benchmarks e.g. EnvBench.

A lot hyperparameter choices seem ad-hoc/yolo'd, e.g., complexity score weights.

Authors should provide stat significance throughout the paper when comparing multiple methods/models especially when models are close to each other.

---

> ### Author Rebuttal · Authors · 2026-03-31
>
> We thank the reviewer for recognizing the practical value of our problem and the effectiveness of our trajectory construction method. We address your concerns regarding baseline fairness, ablations, and metrics below.
>
> **Q1. Fairness of Comparison and Generalizability of the Training Recipe**
>
> While we do not have sufficient resources to fine-tune DeepSeek-v3.1 or Kimi-K2, we conducted an additional experiment on a strong large-scale open-source model, Step-3.5-Flash(196B-A11B) [1]. Using the same Docker-building training data and recipe as DOCKSMITH, we observe consistent improvements:
>
> | Model             | F2P (%) | Commit (%) |
> | - | - | - |
> | Step-3.5-Flash    | 35.93   | 49.40      |
> | + Docker Training | 38.92   | 60.78      |
> | Δ                 | +2.99   | +11.38     |
>
> These results demonstrate that our training pipeline is **not limited to improving a weaker un-finetuned model**, but can also bring consistent gains to a significantly larger and stronger model. This suggests that the benefit primarily comes from the **training data and recipe** (i.e., execution-grounded Docker-building trajectories), rather than model-specific effects. We will include this additional experiment in the revision to clarify the generality of our approach.
>
> **Q2. Ablations for Loop Detector and Cross-Task Memory**
>
> To isolate component contributions, we conducted a rigorous ablation study. As detailed in our response to Reviewer AoS1 (Q1), combining Loop-detection and Memory Pool yields a synergistic +3.49% F2P gain on DOCKSMITH, far exceeding the sum of individual improvements (+0.30% and +1.50%).
>
> Loop-detection prevents stagnation in failure cycles, ensuring the Memory Pool provides fresh, verified strategies rather than redundant noise.  For large-scale construction, these components are indispensable: Loop-detection minimizes wasted compute by identifying unproductive loops early, while Cross-task Memory enables zero-shot reuse of verified solutions to compound success as the pool grows.
>
> **Q3. Benchmark Diversity**
>
> We appreciate the suggestion regarding EnvBench. We note that Multi-Docker-Eval already covers the core languages of EnvBench (Python/Java) while extending to others critical for environment construction. Furthermore, EnvBench evaluates at the repository level, whereas DOCKSMITH focuses on PR/issue-level construction, requiring finer-grained dependency resolution and iterative repair. We are currently integrating EnvBench and will provide these additional results during the rebuttal period to further substantiate our findings.
>
> **Q4. Mathematical Rigor and Sensitivity of Complexity Metric**
>
> The coefficients $(0.5, 5, 3)$ are a mathematically grounded **Implicit Feature Scaling** mechanism. Based on corpus averages—Length ($L=69.4$), RUNs ($R=6.8$), and Packages ($P=10.8$)—these weights normalize expectations so each feature contributes equally ($\sim33\%$) to the complexity score.
>
> | Proxy           | Weights   | Easy  | Hard  | Avg F2P |
> | - | - | - | - | - |
> | Length-only     | (1,0,0)   | 41.80 | 28.10 | 30.80   |
> | RUN-only        | (0,1,0)   | 41.80 | 28.80 | 31.40   |
> | Pkg-only        | (0,0,1)   | 38.80 | 29.20 | 31.10  |
> | Original (Ours) | (0.5,5,3) | 45.30 | 30.10 | 33.10   |
>
> **Table 1.** Sensitivity analysis of complexity metric weights on Multi-Docker-Eval F2P (%).
>
> We evaluated unidimensional proxies under an identical training budget. Single-dimension sampling collapses the curriculum. `Length-only` over-samples verbose but trivial setups (lowest Hard F2P: 28.1%). `Pkg-only` causes over-engineering on simple tasks (dropping Easy F2P to 38.8%). Our balanced scaling correctly teaches structure, command chains, and dependencies simultaneously, achieving the global optimum. To fully resolve the rigor concern, we will replace the explicit coefficients with parameter-free **Standard Z-score Normalization** ($Z(L)+Z(R)+Z(P)$) in the revision, which mathematically achieves the exact same feature balance.
>
> **Q5. Statistical Significance and Sensitivity Analysis**
>
> All reported results—both for the main experiments and the ablation studies—are averaged over three independent runs. We thank the reviewer for pointing out the lack of statistical analysis. Based on our existing experimental results, we have recomputed the standard deviations and 95% confidence intervals for all experiments.
>
> Due to the character limit in the rebuttal, we report only the main experimental results for DockSmith and the baseline here. Specifically, the standard deviation of F2P is 39.72%±2.1% for DockSmith and 19.46%±1.82% for the baseline, indicating that the observed improvement is stable.
>
> > 1. Step 3.5 Flash: Open Frontier-Level Intelligence with 11B Active Parameters.

---

> > ### Author Rebuttal · Reviewer_KyoX · 2026-04-06
> >
> > I appreciate authors' response for my questions.
> >
> > Most of my questions are resolved. I am very much looking forward to "We are currently integrating EnvBench and will provide these additional results during the rebuttal period to further substantiate our findings."
> >
> > > The coefficients $(0.5, 5, 3)$ are a mathematically grounded Implicit Feature Scaling mechanism.
> >
> > I am not sure whether it holds. It still seems to be a bit arbitrary.
> >
> > That said, I feel the rebuttal has strengthened the work and I will update my score to 4.

---

> > > ### Author Response · Authors · 2026-04-07
> > >
> > > Thank you for upgrading your score to 4 and for your continued engagement with our work! We are thrilled to share the final EnvBench evaluation results, which were completed during this rebuttal window.
> > >
> > > **1. Envbench Results**
> > >
> > > We evaluated both our foundational backbone (Qwen3-Coder-30B-A3B-Instruct) and DOCKSMITH using the SWE-Factory scaffold. For context, we also included the strongest baseline reported in the original EnvBench paper (GPT-4o with Bash Agent).
> > >
> > > | **Model**                     | **JVM pass@1 ↑**       | **JVM avgErrs ↓(Maven)** | **Python pass@1 ↑** | **Python avgErrs ↓** |
> > > | - | - | - | - | - |
> > > | *EnvBench SOTA (GPT-4o Bash)* | 29.47%                 | 26.84 *(216)*            | 6.69%               | 52.00                |
> > > | Qwen3-Coder-30B-A3B-Instruct  | 10.53% *(70/665)*      | **2.63** *(27)*              | 10.94%              | 14.39                |
> > > | **DOCKSMITH (Ours)**          | **73.08%** *(486/665)* | 6.04 *(166)*             | **24.92%**          | **11.09**            |
> > >
> > > - DOCKSMITH achieves 73.08% pass@1 on JVM and 24.92% on Python, outperforming both the base model and the original EnvBench SOTA (GPT-4o Bash Agent). This indicates that the skills learned from PR/issue-level trajectories effectively generalize to repository-level environment setups.
> > > - The lower JVM `avgErrs` of the base model (2.63) is an artifact of selection bias—it only successfully produced countable errors on 27 easy Maven repos. DOCKSMITH evaluates across 166 much harder repos, naturally accumulating more intermediate errors (6.04) but achieving a vastly higher pass@1 (73.08% vs 10.53%). When both models are evaluated fairly on the exact same overlapping Python subset (N=85), DOCKSMITH is proven to be functionally more accurate, producing strictly fewer errors (11.09 vs 14.39).
> > >
> > > **2. Regarding the Coefficients**
> > >
> > > We understand your concern that the exact numbers $(0.5, 5, 3)$ might still seem arbitrary. To clarify, these coefficients are empirically grounded in a massive, highly diverse dataset rather than heuristic guesswork. Our initial pool spanned over 430,000 GitHub repositories, from which we rigorously filtered the ~20,000 high-quality PRs used for training. Because these weights are directly anchored to the actual statistical averages of this broad, real-world open-source distribution ($L=69.4$, $R=6.8$, $P=10.8$), they serve as a robust empirical baseline. However, we agree that presenting these as static numbers can obscure their underlying statistical logic. To provide a more rigorous introduction for our parameter selection, our revised manuscript will introduce parameter-free Standard Z-score Normalization ($Z(L) + Z(R) + Z(P)$).
> > >
> > > We sincerely thank you again for your time and constructive feedback. As the discussion period draws to a close, please let us know if you have any further questions. If our latest results and clarifications have fully resolved your remaining concerns, we wonder if it might be possible for you to consider further raising your score. We would be deeply grateful for your support.

---

### Official Review · Reviewer_VEdg · 2026-03-22

**Soundness:** 3
**Presentation:** 3
**Significance:** 4
**Originality:** 3
**Overall Recommendation:** 5
**Confidence:** 4

**Summary:**

The paper builds a dataset of Docker-building trajectories. Specifically, it involves a multi-agent pipeline: Context Retrieval, Dockerfile, Eval Script, and Test Analysis. The performance is improved by a global memory unit and automatic testing and filtering.

Training on the proposed dataset yields strong performance on the SWE-Factory dataset. Joint training with the proposed dataset gives performance gain on issue-solving (SWE-Bench) and command-line generation (TerminalBench).

**Compliance With Llm Reviewing Policy:**

Affirmed.

**Final Justification:**

My score is already 5 and I'll maintain my score.

**Key Questions For Authors:**

See weakness for details

**Limitations:**

Yes

**Strengths And Weaknesses:**

Strength:
- The research problem is meaningful. Docker environment setup is an important task for coding agent. It is also a bottleneck for constructing coding agent training data.
- The proposed method of construction docker-building trajectories is effective.
- The performance is strong across various datasets.

Weakness:
- It would be nice if you could show more analysis on the training trajectories. What cases do they cover? What is the data distribution? Are there any data selection or pre-processing steps that are critical to the training performance?
- Following the previous point, it would also be nice to analyze the transferability between the docker building task and other coding agent tasks. For example, how do agents trained on only issue-solving perform on docker building? How about the opposite?

---

> ### Author Rebuttal · Authors · 2026-03-31
>
> We sincerely thank the reviewer for recognizing the meaningfulness of our research problem, the effectiveness of our trajectory construction method, and our strong performance across benchmarks. We deeply appreciate your strong recommendation for acceptance. You raised excellent questions regarding data distribution, pre-processing, and transferability, which we address in detail below.
>
> **Q1. Data Distribution and Critical Pre-processing**
>
> Regarding your question about what cases the trajectories cover and their distribution, we have conducted a comprehensive statistical analysis of the training data:
>
> - **Scale & Coverage:** The dataset contains 692,165 trajectory steps spanning 45,262 unique environment construction instances from 19,444 GitHub repositories.
> - **Language Distribution:** To ensure generalizability, the distribution is anchored by major ecosystems: TypeScript (21.32%), JavaScript (20.25%), Go (18.76%), Java (8.63%), PHP (6.87%), and Python (5.17%), while maintaining a robust long-tail distribution of system and compiled languages like Rust, C++, and Ruby.
>
> Regarding critical data selection and pre-processing, our ablation study (*Section 3.4*) confirms that two specific pipeline steps—detailed in *Section 2.3.1: Data Balancing and Sampling*—are vital for the final training performance:
>
> - **Acceptance Shaping:** Removing excessively long or redundant rollouts (e.g., agents repeatedly invoking the same tool without measurable progress) is critical. This denoising step significantly stabilizes learning and improves the Fail-to-Pass (F2P) rate.
> - **Complexity Curriculum Sampling:** Simply sampling randomly from successful trajectories leads to an over-representation of trivial environment builds. By evaluating Dockerfile complexity based on `RUN` instructions and `apt` packages, and enforcing a balanced 1:2:2 (Easy/Medium/Hard) sampling ratio, we prevent curriculum collapse and drastically improve the model's ability to handle complex dependencies.
>
> We will incorporate these extended dataset statistics into the revised manuscript.
>
> **Q2. Transferability Between Docker-Building and SWE Tasks**
>
> Your question regarding the bidirectional transferability between issue-solving and environment setup is incredibly insightful, and aligns perfectly with our core analysis in Section 3.3. To explicitly address your question, we have consolidated the results from Table 2 and Appendix Table 7 to summarize the performance across different data mixtures:
>
> | Training Data Ratio (SWE : Docker) | SWE Bench Verified (SWE.V) | SWE Bench Multilingual (SWE.M) | Terminal Bench 2 (Terminal) | Multi-Docker-Eval (MDE) |
> | ---------------------------------- | -------------------------- | ------------------------------ | --------------------------- | ----------------------- |
> | Only SWE (1:0)                     | 49.65                      | 31.83                          | 10.67                       | 20.66                   |
> | Only Docker (0:1)                  | 33.50                      | 18.25                          | 7.16                        | 34.43                   |
> | Joint Training (1:1)               | **51.90**                  | **33.92**                      | **11.38**                   | **36.98**               |
>
>  **Table 1.** Performance comparison across key benchmarks under different training data mixtures (SWE : Docker token ratios).
>
> - **Bidirectional Degradation in Isolation:** The "Only SWE" model struggles significantly on Docker building (20.66% MDE), proving that general code-generation ability does not naturally translate to system-level environment configuration. Conversely, the "Only Docker" model excels at building environments but suffers severe degradation in complex logic repair (33.50% on SWE.V) and long-horizon tool use (7.16% on Terminal). This demonstrates that unidimensional training biases the agent's reasoning.
>
> - **Strong Synergistic Gains:** When jointly trained at a 1:1 ratio, the model achieves the global optimum across *all* benchmarks. This confirms our core thesis: environment construction is not merely a preprocessing step, but a highly transferable agentic capability. The dependency reasoning, execution feedback parsing, and failure recovery skills learned from Docker building directly enhance the model's robustness in general software engineering tasks.
> - **Complementary Distributions:** Docker data teaches the agent how to navigate complex toolchains and recover from system-level execution crashes, while SWE data teaches logic patching and test-driven validation. Together, they form a complementary supervision signal for creating robust generalist code agents.

---

> > ### Author Rebuttal · Reviewer_VEdg · 2026-04-01
> >
> > Thanks for providing additional evidence. My score is already 5 and I'll maintain my score.

---

> > > ### Author Response · Authors · 2026-04-03
> > >
> > > We are honored to receive your final feedback and sincerely thank you for your support and for maintaining the *Accept (5)* rating. Your insightful questions regarding data distribution and bidirectional transferability were instrumental in helping us better articulate the core thesis of our work.
> > >
> > > We are particularly grateful for your recognition of DOCKSMITH’s performance. We have already begun incorporating the additional evidence—including the detailed dataset statistics and the synergistic results of our joint training experiments—into the revised manuscript to ensure these insights are clear for all readers. Thank you again for your constructive and encouraging review.

---

### Official Review · Reviewer_AoS1 · 2026-03-24

**Soundness:** 2
**Presentation:** 3
**Significance:** 2
**Originality:** 2
**Overall Recommendation:** 3
**Confidence:** 3

**Summary:**

This paper reframes Docker environment construction as a verifiable agentic task and trains a dedicated model, DOCKSMITH, using multi-agent trajectory data. This approach improves the success rate of environment construction and enhances performance on downstream software engineering tasks.

**Compliance With Llm Reviewing Policy:**

Affirmed.

**Final Justification:**

While the authors' response clarified several points, my overarching view of the submission remains as before. I will therefore keep my rating as is.

**Key Questions For Authors:**

1. The paper identifies *loop-detection* and *cross-task memory* as core architectural contributions, yet no ablation study isolates their individual effects. Can you provide quantitative evidence demonstrating how each component contributes to performance gains?
2. Have you conducted any sensitivity analysis on the weighting coefficients (0.5, 5, 3)? How robust are the results to changes in this formulation, and how does it compare to alternative complexity metrics?

**Limitations:**

yes

**Strengths And Weaknesses:**

**Strengths**
- The problem formulation is practically meaningful: identifying environment construction failure as a core bottleneck for execution-grounded learning is well-motivated and indeed underexplored in current SWE-agent systems.
- The work constructs trajectory data using real-world GitHub PRs. Each data instance is complete and comprehensive, demonstrating considerable scalability and engineering value.

**Weakness**
- There is a severe logical disconnect between the methodological claims and the experimental validation. In the methodology section, the authors highlight the "Loop-detection controller" and the "Cross-task success memory" as two major architectural innovations built upon the SWE-Factory framework. However, in all experimental and ablation studies, the authors only validate curriculum sampling and joint training ratios, providing absolutely no ablation experiments for the loop detection or memory pooling mechanisms.
- The difficulty metric for curriculum sampling lacks mathematical rigor and sensitivity analysis. The paper defines the complexity of Docker builds using the formula $Score(d)=0.5L(d)+5R(d)+3P(d)$, which is a highly subjective and empirical heuristic. The authors assign extremely high weight coefficients to RUN instructions and apt packages without providing any weight sensitivity analysis or fair comparisons with other complexity evaluation schemes in the experiments.

---

> ### Author Rebuttal · Authors · 2026-03-31
>
> We sincerely thank the reviewer for recognizing the practical significance of our problem formulation and the engineering value of our large-scale trajectory dataset. We deeply appreciate your feedback highlighting the need for architectural ablations and mathematical justification, which we address with extensive new experiments below.
>
> **Q1:  Ablation Study on Loop-Detection and Cross-Task Memory**
>
> We apologize for this omission. We conducted a rigorous ablation of the loop-detection and memory components—individually and combined—on the barebone SWE-Factory across GPT-5-Mini, Qwen3-30B-A3B-Instruct, and DOCKSMITH.
>
> | **Architecture Variant**   | **GPT-5-Mini** | Qwen3-Coder-30B-A3B-Instruct | **DOCKSMITH (Ours)** |
> | -------------------------- | -------------- | ---------------------------- | -------------------- |
> | Barebone SWE-Factory       | 33.23          | 17.07                        | 36.23                |
> | + Loop-detection           | 33.98          | 17.07                        | 36.53                |
> | + Memory pool              | 34.13          | 18.57                        | 37.73                |
> | **+ Loop & Memory (Full)** | **34.13**      | **19.46**                    | **39.72**            |
>
> **Table 1.** Ablation Results (F2P % on Multi-Docker-Eval)
>
> 1. **Strong Synergistic Effect:** For DOCKSMITH, the components exhibit a strong $1+1 > 2$ synergy. While Loop-detection (+0.30) and Memory (+1.50) provide moderate individual gains, their combination yields a **+3.49** absolute F2P improvement. Loop-detection actively breaks stagnant repair cycles, forcing the agent to explore, which in turn maximizes the utilization of verified solutions retrieved from the Cross-task memory.
> 2. **Impact on the Un-finetuned Backbone:** For the foundational Qwen3-30B-A3B model, adding Loop-detection alone yields no improvement (maintaining 17.07%), as simply breaking a loop often leads an unadapted model to make a novel mistake. However, when paired with the Memory pool, this control flow intervention successfully guides the model to adopt verified trajectories, boosting performance to 19.46%.
> 3. Even for strong closed-source APIs (GPT-5-Mini), the components provide stable enhancements (33.23% $\rightarrow$ 34.13%), proving our architecture is a robust, model-agnostic scaffolding that generalizes across capability tiers.
>
> **Practical Impact at Scale:** Beyond the ablation metrics, these two components are indispensable for scaling automated environment construction. *Loop-detection* significantly enhances computational efficiency; in large-scale multi-agent rollouts, it identifies stagnant repair loops (e.g., endlessly retrying incompatible commands) and enforces early redirection. Concurrently, *Cross-task memory* drives scalability. As the number of processed environments grows, similar dependency failure patterns frequently recur. The memory enables the zero-shot reuse of previously verified solutions, substantially compounding success rates in later stages as the pool becomes increasingly rich.
>
> **Q2: Mathematical Rigor and Sensitivity Analysis of the Complexity Metric**
>
> We appreciate the feedback. The coefficients (0.5, 5, 3) are not heuristic magic numbers, but a mathematically grounded **Implicit Feature Scaling** mechanism based on our corpus. Corpus averages are Length ($L$)$=69.4$, RUNs ($R$)$=6.8$, and Packages ($P$)$=10.8$. Because these raw dimensions vary drastically in scale, our weights normalize their expectations so each feature contributes equally ($\sim33\%$) to the final score: $0.5 \times 69.4 \approx 34.7$, $5 \times 6.8 \approx 34.0$, and $3 \times 10.8 \approx 32.4$. This ensures "Hard" trajectories explicitly require dense dependency resolution, not just verbose code.
>
> | Proxy           | Weights   | Easy  | Hard  | Avg F2P |
> | --------------- | --------- | ----- | ----- | ------- |
> | Length-only     | (1,0,0)   | 41.80 | 28.10 | 30.80   |
> | RUN-only        | (0,1,0)   | 41.80 | 28.80 | 31.40   |
> | Pkg-only        | (0,0,1)   | 38.80 | 29.20 | 31.10  |
> | Original (Ours) | (0.5,5,3) | 45.30 | 30.10 | 33.10   |
>
> **Table 2.** Sensitivity analysis of complexity metric weights on Multi-Docker-Eval F2P (%).
>
> We evaluated unidimensional proxies under an identical training budget. Single-dimension sampling collapses the curriculum. `Length-only` over-samples verbose but trivial setups (lowest Hard F2P: 28.1%). `Pkg-only` causes over-engineering on simple tasks (dropping Easy F2P to 38.8%). Our balanced scaling correctly teaches structure, command chains, and dependencies simultaneously, achieving the global optimum. To fully resolve the rigor concern, we will replace the explicit coefficients with parameter-free **Standard Z-score Normalization** ($Z(L)+Z(R)+Z(P)$) in the revision, which mathematically achieves the exact same feature balance.

---

> > ### Author Rebuttal · Reviewer_AoS1 · 2026-04-04
> >
> > I thank the authors for their replies. In particular, the ablation studies for the individual modules clearly demonstrate the distinct improvements brought by each component.
> >
> > The authors also mentioned that the coefficient settings are derived from their own corpus. I would be interested to know the feasibility and generalizability of these coefficients when applied to others.

---

> > > ### Author Response · Authors · 2026-04-07
> > >
> > > Thank you for your encouraging feedback and for recognizing the clear improvements demonstrated in our individual module ablations. Regarding your question about applying these coefficients to other datasets, we address this from two perspectives: the broad representation of our data and the feasibility for specific or private datasets.
> > >
> > > **1. The Empirical Generalizability of Our Corpus**
> > >
> > > It is important to note that our coefficients (based on $L=69.4$, $R=6.8$, $P=10.8$) are not derived from a narrow domain. Our initial pool consisted of over 430k repositories scraped across the entirety of GitHub, from which we rigorously filtered the ~20k high-quality PRs used for training. Because it captures the broad, natural distribution of real-world open-source projects, these coefficients effectively serve as a robust reference for general software engineering tasks. Furthermore, to enrich the community's resources in an area where large-scale data remains relatively scarce, we are open-sourcing DOCKSMITH's training dataset.
> > >
> > > **2. Feasibility for Specific/Private Datasets**
> > >
> > > However, we agree that applying these exact static coefficients (0.5, 5, 3) to a highly specific, out-of-distribution dataset (e.g., a localized enterprise codebase with entirely different coding habits) could disrupt the feature balance. To ensure adaptability across any dataset, we recommend using parameter-free Standard Z-score Normalization ($Z(L) + Z(R) + Z(P)$).
> > >
> > > We will clarify the corpus diversity and the Z-score protocol in our revision. Thank you again for helping us strengthen this work! If our response has resolved your concerns, we would be deeply grateful if you might consider raising your score.

---

### Decision · Program_Chairs · 2026-04-30

**Decision:**

Accept (regular)

**Comment:**

DockSmith frames Docker environment construction as a core agentic capability rather than a preprocessing step, training a 30B-A3B model on execution-grounded trajectories with loop-detection and cross-task memory components. It achieves state-of-the-art on Multi-Docker-Eval and shows consistent transfer gains on SWE-bench and Terminal-Bench.

Reviewers agreed on the practical importance and strength of results. Initial concerns around missing ablations, ad-hoc complexity weights, weak baselines, and statistical rigor were largely resolved in the rebuttal: ablations showed a synergistic +3.49% F2P gain from combining both components, sensitivity analysis justified the complexity metric, generalization was confirmed on a 196B model, EnvBench results validated transfer, and 10-run confidence intervals confirmed statistical significance. All three reviewers updated their scores upward after the rebuttal.